# DoVer: Intervention-Driven Auto Debugging for LLM Multi-Agent Systems

**Ming Ma**[1,2*] , **Jue Zhang**[3†] , **Fangkai Yang**[3†],
**Yu Kang**[3], **Qingwei Lin**[3], **Saravan Rajmohan**[3], **Dongmei Zhang**[3]

[1] Institute of Neuroscience,
  State Key Laboratory of Brain Cognition and Brain-inspired Intelligence Technology,
  Center for Excellence in Brain Science and Intelligence Technology,
  Chinese Academy of Sciences, Shanghai, 200031, China
[2] School of Future Technology, University of Chinese Academy of Sciences,
  Beijing, 100049, China
[3] Microsoft
mam2022@ion.ac.cn, {juezhang, fangkaiyang}@microsoft.com

## Abstract

Large language model (LLM)–based multi-agent systems are challenging to debug because failures often arise from long, branching interaction traces. The prevailing practice is to leverage LLMs for log-based failure localization, attributing errors to a specific agent and step. However, this paradigm has two key limitations: (i) log-only debugging lacks validation, producing untested hypotheses, and (ii) single-step or single-agent attribution is often ill-posed, as we find that multiple distinct interventions can independently repair the failed task. To address the first limitation, we introduce **DoVer**, an intervention-driven debugging framework, which augments hypothesis generation with active verification through targeted interventions (e.g., editing messages, altering plans). For the second limitation, rather than evaluating on attribution accuracy, we focus on measuring whether the system resolves the failure or makes quantifiable progress toward task success, reflecting a more outcome-oriented view of debugging. Within the Magentic-One agent framework, on the datasets derived from GAIA and AssistantBench, DoVer flips 18–28% of failed trials into successes, achieves up to 16% milestone progress, and validates or refutes 30-60% of failure hypotheses. DoVer also performs effectively on a different dataset (GSMPlus) and agent framework (AG2), where it recovers 49% of failed trials. These results highlight intervention as a practical mechanism for improving reliability in agentic systems and open opportunities for more robust, scalable debugging methods for LLM-based multi-agent systems. Project website and code will be available at https://aka.ms/DoVer.

## 1 Introduction

The advancement of Large Language Models (LLMs) has led to the rapid rise of LLM-based agent systems, particularly multi-agent architectures where agents of different roles work collaboratively to solve complex tasks Li et al. (2023); Wu et al. (2023a); Hong et al. (2024). As these systems are increasingly developed and deployed in production, the need to debug their failures becomes inevitable during their lifecycle. Importantly, by "failures" we do not refer to conventional software errors (e.g., exceptions or crashes), but rather to the errors where the system executes without interruption yet produces incorrect or unsatisfactory results Mialon et al. (2024); Yoran et al. (2024). Such failures frequently arise in scenarios where one diagnoses why an agent system underperforms on benchmark tasks during the development phase, or when one addresses user-reported dissatisfaction (e.g., a 'thumbs-down' signal with textual feedback) from an online deployed system.

---

[*] Work is done during an internship at Microsoft.
[†] Corresponding authors.

Debugging failures in LLM-based agent systems presents unique challenges. These systems typically involve multiple rounds of LLM calls, each with extensive textual context, making manual log inspection labor-intensive. Furthermore, in multi-agent tasks, tracking inter-agent information flow is crucial, as failures often stem from *Inter-Agent Misalignment* Cemri et al. (2025). Recent efforts address these issues by using LLMs to analyze system failures Zhuge et al. (2024); Zhang et al. (2025c;a), often via single-agent, single-step failure attribution. In this method, an execution log is input to an LLM tasked with identifying the agent and step responsible for the failure Zhang et al. (2025c). However, as shown in our reproduction study (Section 3), log-based failure attribution is fundamentally limited by the uncertainty of ground-truth annotations. This uncertainty arises for several reasons: agent systems often employ multiple strategies (e.g., ReAct Yao et al. (2023)) with distinct failure points in a single session, and inter-agent misalignment can render the assignment of responsibility to a single agent or step ambiguous.

To circumvent the limitations of uncertain ground-truth attribution, we propose *explicit validation via intervention*, introducing **DoVer** (**Do**-then-**Ver**ify), an intervention-driven framework for automated debugging. DoVer explicitly validates failure attribution hypotheses derived from session logs by intervening at suspected failure points, modifying agent instructions or task plans, while preserving prior context. The system is then re-executed from the intervention point onward. If the failure resolves, the hypothesis is supported; if it persists despite faithful intervention, the hypothesis is refuted. This process enables an iterative cycle of hypothesis generation and validation.

DoVer also supports interventions across multiple steps rather than restricting to single-point edits. By decomposing the failure trace into separate trials, we intervene at each and assess the impact. Within the Magnetic-One (M1) Fourney et al. (2024) agent framework, our experiments show that DoVer recovers 18% and 28% of failures on datasets from AssistantBench Yoran et al. (2024) and GAIA Mialon et al. (2024), respectively. It further enables the validation or refutation of 30–60% of failure hypotheses, depending on task complexity. To demonstrate the generality of DoVer, we apply it to another agent framework, AG2 AutoGen2 (2025), using a different dataset, GSMPlus Li et al. (2024). DoVer again performs effectively, achieving a 49% flip rate on failure cases.

To summarize, our main contributions are: (i) We propose **DoVer**, an intervention-driven framework for automatically debugging failures in LLM-based multi-agent systems; (ii) We identify and analyze the challenges posed by uncertain ground-truth annotations in log-based failure attribution; (iii) We demonstrate experimentally that DoVer not only recovers a significant portion of failure cases but also enables explicit validation and refutation of failure attribution hypotheses.

## 2 RELATED WORK

### 2.1 FAILURE ANALYSIS AND ATTRIBUTION FOR LLM-BASED AGENT SYSTEMS.

LLM-based agent systems exhibit diverse and frequent failure patterns that accumulate along long execution logs. To characterize why and where errors arise, MAST Cemri et al. (2025) catalogs failures across task interpretation, planning, tool/environment interaction, and verification, while evaluation frameworks Zhuge et al. (2024); Arabzadeh et al. (2024) argue that end-to-end pass/fail is too coarse and introduce requirement graphs or task-utility criteria that reveal where progress stalls. Extending this perspective to execution logs, TRAIL Deshpande et al. (2025) creates turn-level traces and a fine-grained taxonomy (reasoning, planning, execution), empirically showing that even strong long-context models struggle at trace debugging.

A parallel line of work seeks *failure attribution*: identifying the earliest decisive step or agent that is responsible for the earliest sufficient cause of failure Zhang et al. (2025c;a). However, these attributions are inferred from logs and remain an *untested hypothesis* unless validated by execution. In DoVer, we treat attribution as a hypothesis to be tested. We apply a targeted edit at the implicated location (message, plan, tool call), and rerun the system, judging success by milestone/utility gains. This places emphasis on verified repair and is consistent with trajectory-aware evaluation advocated in Arabzadeh et al. (2024); Deshpande et al. (2025).

Recent contemporaneous studies[1] further advance failure attribution by introducing reasoning-driven judges Zhu et al. (2025a), abduct–act–predict counterfactual scaffolding West et al. (2025),

---

[1]These works appeared during the final stage of this work preparation or the subsequent peer-review period.

causal-inference formulations Ma et al. (2025), spectrum-based failure attribution approach Ge et al. (2025), hierarchical error-attribution method Banerjee et al. (2025), and graph-guided failure tracing approach Zhang et al. (2025b). Notably, the observation in West et al. (2025) that adding explicit step indices improve attribution aligns with our own prompt-refinement study in Section 3.

Emerging works have also explored specialized failure-tracer models trained on curated success/failure trajectories Zhang et al. (2025a); Kong et al. (2025). These approaches are orthogonal to DoVer and could be incorporated into the initial failure-attribution stage of DoVer to strengthen its diagnostic signal.

## 2.2 Debugging Approaches for LLM-based Agent Systems

Beyond failure analysis and attribution, several systems explore how to *debug* trajectories, i.e., intervention and replay. Human-in-the-loop tools such as AGDebugger Epperson et al. (2025) enable rewind/edit/re-execute with trace visualization, and graph runtimes like LangGraph LangChain (2025) provide checkpoints, interrupts, and "time-travel" branching. These demonstrate that small, targeted interventions often work, but they are manual and hard to scale. The recent AgentDebug Zhu et al. (2025b) work employs an intervention-driven methodology similar to DoVer, although it does not place particular emphasis on multi-agent settings.

From the software-repair perspective, Rahardja et al. (2025) package real agent-system issues into executable environments with failing tests (AgentIssue-Bench) and find low resolution rates for current software engineering agents, highlighting the difficulty of maintaining agent software. Industrial experience at Google similarly evaluates agent-based program repair on production bugs, showing promise but also current limits Rondon et al. (2025). This motivates *auto debugging* that verifies edits within the original run and measures intermediate progress. DoVer does so by choosing a minimal intervention, re-running the trajectory, and scoring milestone/utility gains, in line with Zhuge et al. (2024); Deshpande et al. (2025).

## 3 From Log-Based Attribution to Intervention-Based Debugging

In this section, we surface several subtleties in existing log-based failure analysis that motivate our intervention-based auto-debugging system. We begin by recapping the prevailing task formulation for log-based single-agent/step failure attribution, and then revisit evaluation on an existing benchmark to highlight sources of uncertainty in ground-truth annotation that shape our design.

**Log-based Single-Agent/Single-Step Failure Attribution.** A common setup for log-based failure attribution proceeds as follows. The input is a session log, typically from an LLM-driven multi-agent system, covering the full trajectory from an initial user query to either a final answer or termination caused by system-defined stopping rules (e.g., a maximum number of replanning rounds). The log is a sequence of agent messages organized by turns.[2] The task is to identify the earliest *single* agent and *single* step responsible for the failure. A failure step is defined as a *decisive error*: if one were to replace the agent's incorrect action at that step with a correct one, the trajectory would subsequently succeed. To evaluate this setup, the Who&When (WW) dataset was introduced in Zhang et al. (2025c). In WW, multiple annotators independently reviewed session logs and then reconciled discrepancies through discussion. Reported results show that state-of-the-art LLMs at the time achieved *below 10%* accuracy on failure step attribution.

**Reproduction and Prompt Refinements.** We reproduced the WW step/agent attribution protocol to understand why step-level accuracy is low. Through failure case analysis, we identified two minimal, non-invasive prompt refinements that consistently improve accuracy: (i) adding *explicit step indices* to the failure log, and (ii) embedding a *concise reminder of the annotators' guidance* as instructions. With these refinements, attribution accuracy rises noticeably. For example, on the Hand Crafted category of WW (WW-HC) with the setting of including ground-truth answers and adopting

---

[2] We do not consider logs from asynchronous agent executions in this work.

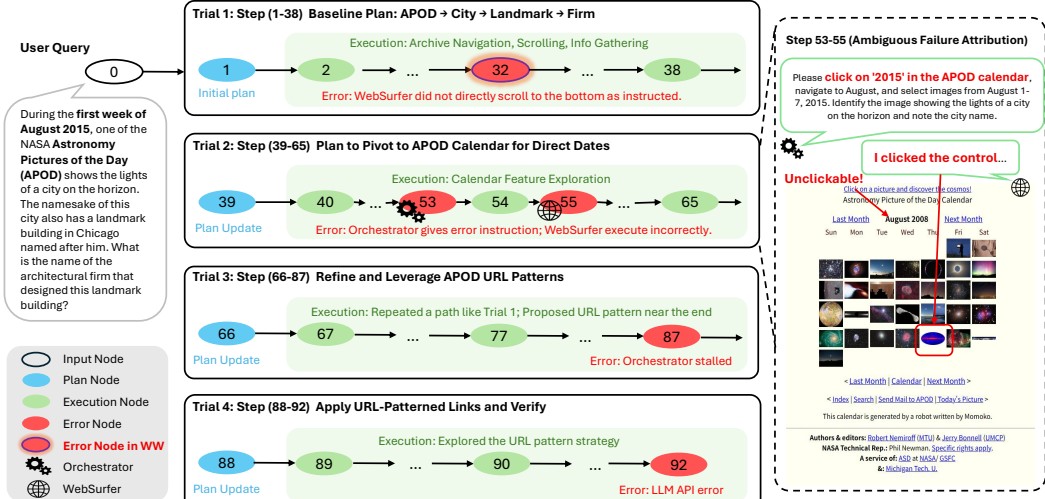

Figure 1: Failure trace of Case 3 in WW-HC, illustrating ambiguity in failure attribution. The session consists of four distinct trials, each initiated by a plan update and executed via a ReAct-style Yao et al. (2023) loop. Different strategies (e.g., direct scrolling in Trial 1 vs. calendar navigation in Trial 2) yield separate error points, making single-step attribution across the session inherently ambiguous. Trial 2 (Steps 53–55) further shows inter-agent misalignment: the Orchestrator issued an invalid instruction, while the WebSurfer compounded the error by executing an unrelated action.

the *All-at-Once* prompt (a single LLM call over the full session log), step attribution accuracy for `GPT-4o` increases from 6% to 24%.[3] Reproduction details are provided in Appendix A.

**Impact of Uncertain Ground-Truth Labels on Attribution.** Despite these gains, absolute step-level accuracy remains low for practical deployment. By comparing model outputs against the ground-truth (GT) labels, we found that uncertainty in the GT annotation is a major contributing factor. Specifically, for the 29 GAIA Mialon et al. (2024) cases in WW, our independent review suggests that 14 of 29 cases exhibit GT uncertainty. This is consistent with WW, which reports annotator uncertainty of 15–30% and an average initial disagreement of ∼20% Zhang et al. (2025c).

To assess the impact of GT uncertainty on model performance, we stratified evaluation into two subsets. For the 14 uncertain cases, average step attribution accuracy is **24%** for `GPT-4o` and **7%** for `GPT-5`. In contrast, for the remaining certain 15 cases, average accuracy increases to **44%** for `GPT-4o` and **53%** for `GPT-5`. These results indicate that GT uncertainty substantially affects model performance. Per-case annotation notes and results are provided in Appendix A.

**Sources of Uncertainty in Failure Agent/Step Annotations.** Our re-annotation surfaced three sources of GT uncertainty among the 14 uncertain GAIA cases in WW:

**(1) Multiple trials within a single session.** Modern agentic systems frequently employ ReAct-style Yao et al. (2023) loops with repeated planning–execution cycles, producing multiple *trials* of task solving. We define a *trial* as the contiguous span starting from a planning step and continuing through the execution steps for that plan. Each trial may contain its *own* decisive error step(s), especially when exploring new strategies or branches. For instance, Figure 1 illustrates the failure trace of Case 3 in WW-HC, which comprises four distinct trials within a single session. Different strategies were explored (e.g., Trial 1 attempted to locate the target webpage via *direct scrolling*, whereas Trial 2 used the *calendar* feature), each introducing distinct potential error points. Consequently, enforcing a single-step attribution across the entire session is intrinsically ambiguous. In our review, 9 of the 14 uncertain cases exhibit this pattern.

---

[3]Unless otherwise specified, `GPT-4o` and `GPT-5` refer to the versions of "GPT-4o-20241120" and "GPT-5-chat-20250807", respectively. All LLM API calls are made through Azure OpenAI using default parameters.

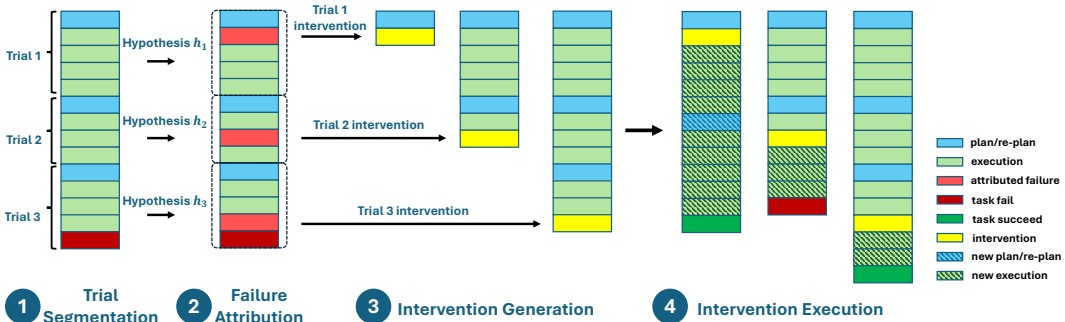

Figure 2: **DoVer** (Do–then–Verify) Debugging Pipeline. (1) *Trial segmentation*: split the failed session log into trials using re-plan steps as cut points. (2) *Failure attribution*: for each trial, propose a hypothesis $h_i$ that marks a faulty step or agent. (3) *Intervention generation*: turn $h_i$ into an actionable intervention that edits either the plan or the attributed message or step in the original log. (4) *Intervention execution*: replay the trajectory *in place*, i.e., preserve all steps before the intervened step, then execute the intervention and measure progress of the new log. Colors indicate plan/re-plan (blue), execution (green), attributed failure (red), terminal failure (dark red), terminal success (dark green), intervention (yellow), new plan/re-plan (blue hatch), and new execution (green hatch).

**(2) Ambiguity from multi-agent coordination.** Attribution to a specific agent/step can be unclear when the underlying issue stems from inter-agent coordination. For instance, when a sub-agent fails to carry out an instruction, responsibility may lie with the sub-agent's capabilities or with ambiguous/misaligned guidance from a supervising agent. In Trial 2 of Figure 1, for example, the Orchestrator agent instructed the WebSurfer agent to click on a non-existent control. Rather than reporting the issue, the WebSurfer instead clicked on an unrelated control that had no connection to the target year "2015". In this case, both agents exhibited failures, making attribution to either one alone inappropriate. Such ambiguous attribution aligns with the *Inter-Agent Misalignment* category identified in Cemri et al. (2025). We observed this phenomenon in 5 of the 14 uncertain cases.

**(3) Cross-annotator alignment challenges.** Even with adjudication, achieving fully aligned labels can be difficult. For 7 of the uncertain cases in WW, we found no clear error at the GT-designated step, suggesting that differing interpretations can persist despite careful protocol design.

**Implications for Intervention-Based Debugging.** The above analysis suggests that *log-only* failure attribution can fundamentally suffer from the issue of uncertain ground-truth labels. Consequently, we propose *explicit validation via intervention*: hypothesize the failure step, *intervene* on it (e.g., replace the action or instruction), and verify whether the trajectory subsequently succeeds. This protocol operationalizes the "decisive error" definition while simultaneously eliminating dependence on noisy human labels and thus reducing annotation burden. Moreover, two design takeaways follow from our uncertainty study. **(i) Trial awareness.** Because logs often contain multiple planning–execution *trials*, interventions must be applied at the trial level, motivating our *trial-based intervention* framework. **(ii) Role-specific interventions.** Given ambiguity in attributing responsibility between an orchestrator and its sub-agents, we adopt a clear taxonomy of interventions: (a) interventions on the *orchestrator's plan* and its *instructions to sub-agents*, and (b) direct capability improvements for sub-agents (e.g., skills). These implications motivate our *intervention-based auto debugging* system; in the next section, we detail the framework and its concrete instantiations.

## 4 METHODOLOGY

Following the implications for intervention-based debugging, we present **DoVer**, a *do-then-verify* debugging pipeline that turns failure-attribution hypotheses into controlled edits and checks whether those edits change outcomes. As shown in Figure 2, DoVer consists of four stages: (1) *trial segmentation* to break an execution log into trials, (2) *hypothesis generation* to hypothesize a failure step or agent, (3) *intervention generation* to synthesize a testable change, and (4) *intervention execution* with differential evaluation against the original run using task success and a progress score.

### 4.1 DoVer Pipeline

**Trial Segmentation.** After a task is executed by an agentic system, we have obtained a long execution session log $\tau = \{(a_t, m_t, \sigma_t)\}_{t=1}^{T}$, where $a_t$ represents the active agent that produces the message $m_t$ at step $t$, and $\sigma_t$ keeps stateful information that is necessary for state restore and replay. For example, this includes historical context (e.g., prior messages sent to LLMs) as well as browsing history for agents acting in a web-browsing role. Since modern LLM-based agent systems Fourney et al. (2024); SmolAgents (2025); AutoGen2 (2025) have the self-reflection capabilities Yao et al. (2023); Shinn et al. (2023), and re-plan after reflection, we then segment the trace $\tau$ into trials $\tau^i$ using re-plan steps as segmentation point, as illustrated in Figure 2. This trial segmentation shortens context so LLMs can reason about a single causal chain, and enables independent and parallel interventions for efficiency. Thus, multiple hypotheses can be proposed for a single session trace, which is essential as many failures admit more than one viable repair.

While one could implement trial segmentation by leveraging system-specific message patterns (e.g., in M1, plan or re-plan steps often begin with "*We are working to address the following user request*"), we instead adopt a prompt-based approach. Specifically, we employ LLMs to reason over the full session log and identify planning-related steps. This method generalizes more effectively to other agent systems whose log patterns for planning steps are not known a priori. The prompt used for trial segmentation is provided in Figure 5 in Appendix B.

**Failure Attribution.** For each trial $\tau^i$, we generate candidate failure attribution hypothesis

$$h_i = (\hat{a}_{\hat{t}}^i, r_{\hat{t}}^i),$$

where $\hat{t}$ is the step index of attributed failure step, $i$ is the trial index, $\hat{a}^i$ the suspected agent, and $r^i$ a natural-language rationale. We build on existing log-based attribution methods (e.g., our improved All-at-Once prompt from Section 3) but crucially do not require perfect precision, since correctness will later be tested via explicit intervention. Here we adapt the All-at-Once prompt (see Figure 6 in the Appendix B) so that it can be applied to session logs segmented into trials by the preceding step.

**Intervention Generation.** In this stage, failure attribution hypotheses are transformed into concrete interventions. Each intervention $I_i$ represents a targeted edit to the failing context, informed both by the specific failure hypothesis and by the broader context of the intervention step. As noted in Section 3, ambiguity may arise in attributing responsibility between the orchestrator and its sub-agents. We thus distinguish interventions directed at orchestrator from those directed at sub-agents.

To keep our method more agnostic to the underlying agent architecture, we focus primarily on interventions at the orchestrator level, i.e., interventions applied at the message-passing layer via direct edits to agent messages. This choice improves generality and simplicity but introduces a trade-off: we cannot directly intervene on sub-agents to enhance their capabilities, which would typically require substantial system-level modifications (e.g., extending the WebSurfer agent to support in-page search). In summary, the interventions considered in this work fall into two categories:

- *Modified Instructions to Sub-Agents:* Adjusting the orchestrator's messages to sub-agents to clarify intent, correct arguments, or supply missing context. This approach indirectly influences sub-agent behavior.
- *Plan Updates:* Revising the orchestrator's high-level plan, e.g., reordering, decomposing, or replacing steps, to route around the identified failure.

The prompt used for intervention generation is provided in Figure 8 in Appendix B.

**Intervention Execution.** The agentic system replays each trial with interventions applied *in-situ* at the suspected failure step. All steps are preserved before the intervened step and execute the intervention $I_i$, yielding a counterfactual trace $\tilde{\tau}_I = \{\tau_1, \tau_2, \cdots, \tau_{i-1}, \tilde{\tau}_i\}$. Note that one intervention creates one new trace, verifying the failure attribution of each trial.

### 4.2 Evaluation Metrics

We consider two sets of evaluation metrics to address our research questions: (1) whether failed cases can be turned into successful ones through intervention and (2) how effective our debugging system is at validating or refuting the initial failure hypothesis. To reduce the impact of execution randomness (e.g., LLM stochasticity), we perform *three* independent runs for each intervention.

**Metrics for Turning Failures into Successes.** To answer the first question, we introduce two failure-flipping related metrics: *Trial Success Rate* and *Progress Made*. The *Trial Success Rate* is defined as the ratio of trial runs that successfully complete the task after intervention. The *Progress Made* metric captures whether intervention brings a failed trial closer to success, even if it does not ultimately succeed. Thus, it provides a more fine-grained measure of improvement when the intervention makes the failed case "more correct".

Specifically, when an intervention does not yield full success and human-annotated solution steps are available (e.g., in the GAIA benchmark Mialon et al. (2024)), we could evaluate the degree of progress by comparing the new execution trace against human-annotated steps. For each log $\tau$, we extract up to $K \leq 5$ milestones $\{\mathbf{m}^k\}_{k=1}^{K}$ and measure how many of these milestones the new trace accomplishes. Both the extraction of milestones and their evaluation are performed using LLMs with carefully designed instructions. The actual prompts are provided in Figures 9 and 10 in Appendix B. We then define the milestone achievement count for a trace $\gamma$ as:

$$A(\gamma) = \sum_{k=1}^{K} \mathbb{I}\big[\text{milestone } \mathbf{m}^k \text{ is achieved in } \gamma\big].$$

We measure *Progress Made* as the ratio of additional milestones achieved:

$$Prog(\tau \to \tilde{\tau}_I) = \frac{A(\tilde{\tau}_I) - A(\tau)}{K} \in [-1, 1],$$

where $\tilde{\tau}_I$ is the new execution trace after intervention, $K$ is the number of extracted milestones from human-annotated trace.

In settings where human-annotated intermediate steps are unavailable, the progress metric above cannot be directly applied. In such cases, one may instead directly compare execution traces before and after intervention with LLM-as-a-judge. We leave this alternative evaluation for future work.

**Metrics for Validating Failure Hypotheses.** To evaluate the effectiveness of our debugging system in validating or refuting the initial failure hypothesis, we classify each trial after intervention into four categories: *Validated*, *Partially Validated*, *Refuted* and *Inconclusive*. The *Inconclusive* category is necessary because we frequently observe that agents fail to follow the intervened instruction, resulting in unsuccessful trials. In such cases, it is unclear whether the outcome stems from an incorrect failure hypothesis or from other limitations of the system that prevent the intervention from being carried out.

To handle this ambiguity, we conduct a comparative analysis of traces before and after intervention. We leverage LLMs to determine whether the intervention was faithfully executed by the agent, resulting in a boolean metric "*is_intervention_fulfilled*" for each trial run. The prompt employed in this work can be found in Figure 11 in the Appendix B. Together with the overall *Trial Success Rate* and *Progress Made* metrics, we define the four outcomes as follows:

- *Validated*: At least 2 of 3 repeated runs succeed.
- *Partially Validated*: Fewer than 2 of 3 runs succeed, and at least 2 of 3 runs both fulfill the intervention and show additional 20% progress (i.e., they advance by one key milestone).
- *Refuted*: Same as "Partially Validated," except that progress does not exceed 20%.
- *Inconclusive*: All other cases.

In summary, these metrics allow us to quantify both the extent to which interventions make failed cases more correct and how effectively the system validates or refutes hypotheses. We now apply them in the next subsection to analyze our experimental results.

## 5 EXPERIMENT RESULTS

### 5.1 EXPERIMENT SETUP

**Agent System.** We consider two distinct agent frameworks in this work. Following Zhang et al. (2025c), we begin by conducting experiments using Magentic-One (M1) Fourney et al. (2024), a

popular LLM-based multi-agent framework that was also used for collecting failure traces in WW. We enable DoVer on M1 by adapting the manual debugging tool AGDebugger Epperson et al. (2025) so that DoVer can (without human intervention): (i) save the state at each step as a checkpoint; (ii) load checkpoints from a prior failed run; (iii) intervene by modifying an agent message at a specified step; and (iv) resume execution from the intervened step.

To evaluate DoVer's generality, we further construct a MathChat multi-agent system using a second framework, AutoGen2 (AG2) AutoGen2 (2025); Wu et al. (2023b). This MathChat system is instantiated using the prompts provided in MAST Cemri et al. (2025), and we extend AG2 with checkpointing and re-execution capabilities analogous to those in M1. Implementation details and practical lessons for reducing the integration burden on new frameworks are provided in Appendix C.

**Datasets.** We first follow Zhang et al. (2025c) and include all cases in the *Hand Crafted* category of WW: 28 cases from AssistantBench Yoran et al. (2024) and 29 cases spanning all three GAIA levels Mialon et al. (2024). To increase data volume, we additionally include all 53 Level-1 cases from GAIA's validation set; after excluding those already present in WW, this yields 45 extra cases. We refer to these three sets as *WW-AB*, *WW-GAIA*, and *GAIA-Level-1*, respectively.

For the MathChat system, following MAST Cemri et al. (2025), we additionally use the GSMPlus dataset Li et al. (2024). Concretely, we adopt the 2,400 examples in the "testmini" split and re-collect execution traces with checkpoints for all problems, forming the *GSMPlus* setting used in our AG2-based experiments.

**Failure Trace Collection.** We begin with an initial run over all cases and evaluate outcomes to identify failure traces. The failure logs published with WW and MAST are not directly usable for our purposes: logs of agent messages alone are insufficient to support replay and targeted intervention. The number of failed cases per dataset is given in Table 1 ("Failed Cases" column). The overall success rate on the full GAIA Level-1 set matches the value reported for Magentic-One Fourney et al. (2024), suggesting that our collected failure traces faithfully reflect M1's capabilities. As in WW and MAST, we use `GPT-4o` both to generate traces and to power the intervened session runs.

|          | Total | Failed Cases | Intervened Cases | Intervened Trials | Trials per Case |
|----------|-------|--------------|------------------|-------------------|-----------------|
| WW-AB    | 28    | 26           | 23               | 72                | 3.1             |
| WW-GAIA  | 29    | 26           | 25               | 99                | 4.0             |
| GAIA-Level-1 | 45 | 26           | 25               | 63                | 2.5             |
| GSMPlus  | 2400  | 214          | 141              | 198               | 1.4             |

Table 1: Summary of failed and intervened cases across datasets, showing the total number of cases, failed cases, intervened cases, total intervened trials, and the average number of trials per case.

**Auto Debugging with DoVer.** We apply DoVer to each failed trace using `GPT-4o` while obtaining the progress made metric and failure hypothesis validation results with `GPT-5`. As described in Section 4, DoVer first segments the full trace into distinct trials, then performs failure attribution and intervention generation for each trial. Finally, it collects the intervened session traces and conducts comparative analysis. Table 1 reports the number of cases for which an intervention was successfully generated (LLMs may occasionally conclude, incorrectly, that no mistake occurred), the total number of intervened trials, and the average number of intervened trials per case. On average, we perform about 3 (1.5) intervened trials per case in the AB and GAIA (GSMPlus) datasets, indicating that most cases contain multiple trials and multiple potential failure points, making them worthwhile targets for debugging.

## 5.2 Quantitative Evaluation Results

Table 2 presents the experimental results for the failure-flipping metrics. For making failure cases more correct, the *Trial Success Rate* across all intervened trials is 17.6% for cases in WW, compared to a higher success rate of 27.5% for GAIA-Level-1 cases. This difference can be explained by the fact that WW contains more challenging cases (e.g., Level-2/3 GAIA tasks). A similar pattern is observed for the *Progress Made* metric: interventions yield a 15.7% improvement (i.e., nearly one key milestone) in GAIA-Level-1, but considerably less progress in WW-AB and WW-GAIA.

|  | Intervened Trials | Trial Success Rate | Progress Made |
|---|---|---|---|
| WW-AB | 72 | 17.6% | +0% |
| WW-GAIA | 99 | 17.6% | +8.8% |
| GAIA-Level-1 | 63 | 27.5% | +15.7% |
| GSMPlus | 198 | 49.0% | - |

Table 2: Experimental results on failure-flipping metrics across settings. The table reports the number of *Intervened Trials*, the *Trial Success Rate*, and the average *Progress Made*.

Notably, for WW-AB, almost no progress is achieved after intervention, suggesting that in some situations interventions may even hinder progress toward success. Finally, in the GSMPlus setting, DoVer achieves nearly a 50% trial success rate, underscoring that its effectiveness generalizes well across datasets and agent frameworks. Note that the *Progress Made* metric cannot be computed for GSMPlus due to the absence of human-annotated solution steps in the dataset.

|  | Intervened Trials | Validated | Inconclusive | Partially Validated | Refuted |
|---|---|---|---|---|---|
| WW-AB | 72 | 11 (15.3%) | 48 (66.7%) | 3 (4.2%) | 10 (13.9%) |
| WW-GAIA | 99 | 16 (16.2%) | 57 (57.6%) | 5 (5.1%) | 21 (21.2%) |
| GAIA-Level-1 | 63 | 22 (34.9%) | 18 (28.6%) | 8 (12.7%) | 15 (23.8%) |

Table 3: Validation outcomes of failure hypotheses across datasets. The table reports the number and percentage of trials classified as *Validated*, *Inconclusive*, *Partially Validated*, or *Refuted*.

Turning to the validation of failure hypotheses, Table 3 shows that for both WW-AB and WW-GAIA, the proportions of *Validated* and *Refuted* hypotheses are similar, each around 15%, while the majority (about 60%) fall into the *Inconclusive* category. In contrast, GAIA-Level-1 exhibits higher rates of both validated and refuted hypotheses, with inconclusive cases reduced to about 30%. This pattern suggests that the more difficult cases in WW make it harder for the agent system to reliably carry out the intended interventions.

## 5.3 ABLATION STUDY

**Impact of Different DoVer Underlying Models.** To test whether DoVer depends on a proprietary frontier model, we vary its debugging model while keeping failure traces, the agent system, and all prompts fixed. In the WW-GAIA setting, we replace `GPT-4o` with two locally hosted open-source models, `Qwen3-8B` and `Qwen3-32B` in thinking mode. As shown in Table 4, `Qwen3-8B` recovers 11.3% of 77 trials and Qwen3-32B recovers 16.9% of 87 trials, compared to 17.6% for `GPT-4o` over 99 trials. These results show that (i) DoVer does not rely on a single proprietary backend and works with medium-sized local models, and (ii) larger open-source models (e.g., `Qwen3-32B`) substantially narrow the gap to `GPT-4o`, underscoring DoVer's generality and practicality for open-source deployments.

| DoVer Model | Intervened Trials | Trial Success Rate |
|---|---|---|
| `Qwen3-8B` (0-shot) | 77 | 11.3% |
| `Qwen3-8B` (**3-shot**) | 77 | 14.3% |
| `Qwen3-32B` (0-shot) | 87 | 16.9% |
| `GPT-4o` (0-shot) | 99 | 17.6% |

Table 4: Ablation of DoVer models and few-shot prompting in the WW-GAIA setting.

**Effect of Few-Shot Prompting for Smaller DoVer Models.** To assess whether prompt-based guidance can mitigate the limitations of smaller models, we compare DoVer performance with and without few-shot examples. In the WW-GAIA setting described above, we enhance the intervention-generation prompt for the `Qwen3-8B` model by adding three manually curated few-shot examples, each demonstrating how the orchestrator refines an initially suboptimal instruction into a clear and effective one, along with brief context before and after the intervention. The resulting trial success rate is reported as "`Qwen3-8B` (3-shot)" in Table 4. The results show that `Qwen3-8B` with this

3-shot prompt achieves a 14.3% trial success rate, compared to 11.3% in the zero-shot setting. This improvement indicates that even small models can benefit considerably from lightweight in-context supervision, suggesting that richer prompt design or future supervised or reinforcement learning on intervention data may further narrow the gap to larger models.

**Compare with other Self-Improvement Methods.** To contextualize DoVer's gains, we compare against two self-improvement style methods adapted from Self-Refine Madaan et al. (2023) and CRITIC Gou et al. (2023) in the WW-GAIA setting. In the Self-Refine-style baseline, the underlying LLM first critiques the final answer given the full session log and then generates a revised answer in a second pass. In the CRITIC-style baseline, we similarly elicit feedback from the final answer and log, but inject this feedback as an additional agent message and allow the agent system to run for one extra round so that all tools and sub-agents can react to it. Across all 26 failed WW-GAIA cases, neither baseline is able to flip any failure into success (*0% recovery*), whereas DoVer recovers 17.6% of trials (Table 2). Further examination reveals why these self-improvement methods fall short: trial trajectories between the initial failure point and the final answer are often long, noisy, and highly divergent, making end-of-trace refinement insufficient for reliably redirecting the system. In contrast, DoVer performs *in-situ* interventions at potential failure points, enabling timely and targeted corrections that are essential in multi-agent settings.

**Qualitative Analysis.** Detailed case vignettes and tool enhancement analysis are provided in Appendix D and E .

## 6 CONCLUSION

We introduced **DoVer**, an intervention-driven framework that reframes debugging in LLM-based multi-agent systems as a *do-then-verify* process. Rather than relying solely on log-based attribution, DoVer operationalizes debugging by formulating and directly testing failure attribution hypotheses via targeted interventions. Across datasets from AssistantBench and GAIA under M1, DoVer recovers 18–28% of previously failed trials, and it recovers 49% of failures in GSMPlus when applied within a different agent framework (AG2). Beyond these recoveries, DoVer consistently produces measurable improvements and verifies or falsifies most hypotheses. This outcome-oriented perspective highlights intervention as a practical and scalable tool for improving reliability, while reducing dependence on ambiguous human annotations. By bridging failure analysis with practical repair, DoVer takes a step toward more robust, verifiable, and self-improving multi-agent systems.

## 7 LIMITATIONS AND GENERALIZABILITY

Our empirical evaluation provides evidence of feasibility across two frameworks, though the covered tasks and model families represent only a subset of deployed agentic systems. Consequently, we do not evaluate on long-running production workloads or settings with strict safety-critical requirements.

Methodologically, the two core components of DoVer, trial segmentation and intervention-based validation, are conceptually general, but our current instantiation imposes several preconditions. First, the agent framework must expose sufficiently rich interaction logs and a checkpoint/replay interface so that we can reconstruct contiguous planning–execution trials and splice in edited messages. Integrating DoVer into systems without such facilities (e.g., asynchronous or black-box orchestrators) would require additional instrumentation or higher-level abstractions over the interaction history, and our experience in M1 and AutoGen2 shows that adding checkpointing still requires non-trivial engineering effort. Second, in this work we restrict interventions to the orchestrator's textual messages; we neither modify sub-agent code nor synthesize new tools, which limits our ability to repair failures rooted in missing capabilities rather than mis-specified coordination. Finally, part of our analysis (milestone progress and "intervention fulfilled" labels) relies on LLM-as-a-judge assessments, which may introduce biases despite careful prompt design. Exploring broader domains and architectures, richer intervention spaces (e.g., tool augmentation or code changes), and more human-grounded evaluation protocols are potential directions for future work.

ACKNOWLEDGEMENTS

We thank the anonymous reviewers for their constructive comments and suggestions, which helped improve the clarity and quality of this work. We also thank Tianming Yang, Yanjie Gao and Anyan Chen for helpful discussions. We further thank the creators and maintainers of GAIA, AssistantBench, GSMPlus, Magentic-One, AutoGen2, AGDebugger, and related open-source tools and benchmarks for making their datasets and systems publicly available, which enabled the experiments in this paper.

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

## A    REVISIT LOG-BASED FAILURE ATTRIBUTION ON THE WHO&WHEN DATASET

In this section, we revisit the log-based failure attribution task on the Who&When (WW) dataset. We present the details of our reproduced evaluation on WW as well as two minimal, non-invasive prompt refinements. We further examine the ground-truth labels in WW, providing detailed annotation notes for each examined case and discussing sources of uncertainty during ground-truth annotation.

Our reproduced evaluation on WW focuses on the All-at-Once (AAO) method, in which a single LLM call is made over the full input log. While we also reproduced results for the other two methods, "Step-by-Step" and "Binary Search" introduced in WW, we focus on AAO because it requires only one LLM call and achieves performance comparable to the other methods. In our reproduction, we run experiments with both GPT-4o and GPT-5, repeating each run three times to

| Method | Hand Crafted | | Algorithm Generated | |
|---|---|---|---|---|
| | **Agent-Level Acc.** | **Step-Level Acc.** | **Agent-Level Acc.** | **Step-Level Acc.** |
| Random | 12.00 | 4.16 | 29.10 | 19.06 |
| Baseline (WW) | 55.17 | 5.26 | 54.33 | 12.50 |
| Baseline (GPT-4o) | 55.17±8.09 | 6.04±2.23 | 55.16±2.71 | 15.28±2.70 |
| + Step Index (GPT-4o) | 52.30±1.00 | 20.69±2.98 | 58.73±3.46 | 40.47±4.20 |
| + Guidance (GPT-4o) | 59.19±1.99 | 23.56±4.98 | 57.41±1.83 | 35.45±3.58 |
| + Guidance (GPT-5) | **59.19±1.99** | **23.56±1.00** | **62.43±1.66** | **45.77±1.65** |

Table 5: Reproduced evaluation results on the WW dataset using the All-at-Once method with the ground-truth annotation. Results are reported for both Hand-Crafted and Algorithm-Generated scenarios at agent-level and step-level accuracy. Rows show baseline results from WW, our reproduction with GPT-4o, and refinements with explicit step indices and guidance reminders, as well as the latest GPT-5 model.

reduce randomness. Detailed results are shown in Table 5. Following WW, we include the ground-truth annotation in the AAO prompt (the "With Ground-Truth" setting); results without ground-truth are similar and omitted for brevity.

Table 5 first lists the "Random" setting and the reported results in WW (denoted as "Baseline"). Examining the baseline failure cases yields two observations. *First*, for some cases the predicted failure step index is hallucinated, i.e., the index exceeds the total number of steps in the log. Specifically, we find that the percentages of cases exhibiting this issue are 13.8% and 20.6% for the "Hand Crafted" and "Algorithm Generated" scenarios, respectively. To mitigate this, we add *explicit step indices* to the failure log (see Figure 3). The corresponding results appear in the "+Step Index" row of Table 5. This simple change substantially improves step-level attribution accuracy. We additionally verify that no outputs have out-of-range indices after this modification. *Second*, the baseline AAO prompt does not explicitly restate the guidance used during ground-truth annotation, which may cause LLMs and human annotators to apply different criteria for the failure agent/step. To align the criteria, we embed *a concise reminder of the annotators' guidance* into the baseline prompt (see Figure 3). With this addition, attribution performance improves further; see the "+Guidance" row in Table 5.

Despite these prompt refinements, step-level attribution accuracy remains only around 20%, even with the latest GPT-5 model. To better understand this gap, we compare model outputs with ground-truth labels and find that a major contributing factor is uncertainty in the ground-truth annotation. As discussed in Section 3, we identify three sources of annotation uncertainty. Table 6 provides detailed annotation notes for each GAIA case presented in WW. For example, for Case 3, we list the step ranges for all 4 trial presented in the log. In each trial, we also provide the annotation notes for key steps (e.g., for the current ground-truth annotation Step 32, we agree that it can be a potential error step). Brief trial summary for later Trials 2/3/4 are also given in which we explicitly point out whether new strategies of solving task are explored. Note that we exclude Case 25 due to ambiguity in its question (i.e., two different ways of interpreting the year of 2019).

In our independent annotation, we also tag each case along the following dimensions: (i) whether the ground-truth failure step appears correct; (ii) whether alternative failure steps can be justified when the log contains multiple trials exploring different strategies; (iii) whether agent/step attribution is ambiguous; and (iv) whether API errors or flaky behavior are present. We then mark a case as *uncertain* if any of the first three tags is positive. The full tagging results are summarized in Table 7, which also includes auxiliary information such as the number of trials, the GT failure step from WW, and the outputs from each model run. Outputs matching the GT failure step are highlighted in green. As shown, agreement between model outputs and GT labels is substantially higher for cases with uncertain GT annotation (i.e., where the "Is Current GT Uncertain?" column equals 1) than for cases with more certain GT annotation. This finding indicates that current log-based failure attribution can be confounded by uncertainty in ground-truth labels, motivating the intervention-based auto-debugging approach presented in the main text.

**Optimal Prompt Template**

You are an AI assistant tasked with analyzing a multi-agent conversation history when solving a real world problem.

Here is the Annotation Guideline:

Failure Responsible Agent:

a) Select the single agent that should be directly responsible for this failure in your mind. Allow for some subjectivity, but be prepared to give your reasons.
b) Don't be too strict. If there exist agents that do redundant steps and agents that make mistakes, choose the agent who makes mistakes.
c) If there are no agents that make obvious mistakes, decide one single agent in your mind.
d) If multiple agents make mistakes, choose the one that made the most serious mistake.

Decisive error step:

a) First decide one single mistake agent, then decide one single mistake step. The Mistake step must be made by the mistake agent.
b) If the mistake agent makes mistakes in multiple steps, choose the first step.
c) Index from 0.
Failure Reasons:
a) First, use natural language to describe the reason. E.g.,
"The agent wrote the wrong code."
b) Make sure the reader could understand the annotations.

Others:

a) Accurately record the time of labeling.
b) Mark all annotation if you have any uncertain, and then we need to vote and discuss later.

While following the Annotation Guideline above, you must still strictly produce the final answer in the exact output format specified below.

The problem is: {problem}
The Answer for the problem is: {ground_truth}
Identify which agent made an error, at which step, and explain the reason for the error. Here's the conversation:

[Step {step_idx_i}] {agent_name_i}: {content_i}…

Based on this conversation, please predict the following:
1. The name of the agent who made a mistake that should be directly responsible for the wrong solution to the real world problem.
If there are no agents that make obvious mistakes, decide one single agent in your mind. Directly output the name of the Expert.
2. In which step the mistake agent first made mistake. For example, in a conversation structured as follows:
{
    [Step 0] "agent a": "xx",
    [Step 1] "agent b": "xxxx",
    [Step 2] "agent c": "xxxxx",
    [Step 3] "agent a": "xxxxxxx"
},
each entry represents a 'step' where an agent provides input. The 'x' symbolizes the speech of each agent. If the mistake is in agent c's speech, the step number is 2. If the second speech by 'agent a' contains the mistake, the step number is 3, and so on.
Please determine the step number where the first mistake occurred.
Important: Count steps strictly using the bracketed indices [Step k] shown in the conversation/example, and report the first [Step k] where the mistake first appears.
3. The reason for your prediction.
Output exactly in this format (no extra text):
Please answer in the format:
Agent Name: (Your prediction)\n
Step Number: (Your prediction)\n
Reason for Mistake: \n

Figure 3: Extended prompt template: orange marks explicit step indices, and blue marks the embedded concise reminder of annotators' guidance.

Table 6: Detailed annotation notes for each GAIA case in WW. The table records trial-level observations, potential errors, ambiguous attributions, and API issues associated with the ground-truth labels.

| WW ID | Annotation Notes |
|---|---|
| 3 | **Trial 1** (Step 0–38):
– Step 32: current GT; potential error as WebSurfer did not directly scroll to the bottom as instructed.
**Trial 2** (Step 39–65): Tried a **new** strategy using the calendar feature but remained stuck navigating within the calendar.
– Steps 53–55: **Ambiguous Agent/Step Attribution**; WebSurfer could not click the target year instructed by Orchestrator because the calendar tool lacks a year filter.
**Trial 3** (Step 66–87): Repeated a path similar to Trial 1, proposing a **new** strategy of leveraging URL patterns near the end.
**Trial 4** (Step 88–92): Explored the URL-pattern strategy but encountered an LLM API error.
– Step 92: LLM API error due to content filter. |
| 5 | Step 12: current GT; potential error due to missing OCR text.
Steps 12–16: **Ambiguous Agent/Step Attribution**; Orchestrator did not verify the OCR output but still instructed WebSurfer to proceed, and WebSurfer did not perform a sanity check and guessed the answer. |
| 9 | **Trial 1** (0–25):
– Step 25: current GT; potential error due to unnecessary replanning.
**Trial 2** (26–51): Repeated the strategy in Trial 1.
**Trial 3** (52–74): Tried a **new** website.
**Trial 4** (75–94): Visited the same websites as before. |
| 11 | **Trial 1** (Step 0–38):
– Step 24: current GT; no obvious error. If attributing a mistake to not recognizing clickable tabs, one may attribute earlier Steps such as 12/16/20.
**Trial 2** (39–73): Repeated the same page-scrolling as Trial 1; tried a **new** strategy of in-page search but got stuck.
– Step 67: potential error using the email feature for search.
**Trial 3** (74–115): **Made progress** and identified the correct oldest flavor; stuck at inspecting the background photo.
**Trial 4** (116–129): Tried directly searching for the target photo from **other** sources; terminated due to max rounds reached.
– Step 129: almost solved the task but stopped due to max rounds reached. |
| 20 | **Trial 1** (0–34):
– Step 3: current GT; no obvious error.
– Step 24: potential error; unable to open the downloaded PDF.
– Steps 30–32: **Ambiguous Agent/Step Attribution**; Orchestrator asked FileSurfer to open a paper that had not been downloaded; FileSurfer tried to open a hallucinated file path.
**Trial 2** (35–66): Still stuck downloading/opening the PDF; LLM API error due to content filter.
– Step 66: LLM API error due to content filter. |
| 21 | Step 4: current GT; no obvious error.
Step 24: LLM API error due to content filter. |
| 22 | Step 4: current GT; no obvious error.
Steps 14–20: **Ambiguous Agent/Step Attribution**; Orchestrator instructed FileSurfer to process a downloaded PDF that WebSurfer had not clearly downloaded, resulting in "File not found."
Step 23: LLM API error due to content filter. |

*Continued on next page*

*Continued from previous page*

| WW ID | Annotation Notes |
|---|---|
| 27 | **Trial 1** (Step 0–30):
– Step 4: current GT; no obvious error.
– Steps 18–20: **Ambiguous Agent/Step Attribution**; Orchestrator instructed FileSurfer to process a downloaded PDF that WebSurfer had not clearly downloaded, resulting in "File not found."
**Trial 2** (Step 31–50): Encountered the same "File not found" issue, tried to resolve it, then terminated due to LLM API error.
– Step 50: LLM API error due to content filter. |
| 41 | **Trial 1** (0–37):
– Step 8: current GT; no obvious error.
– Step 16: potential error; access to the desired website was blocked by human verification.
**Trial 2** (38–82): Tried a **different** website; attempted a **new** strategy of asking for help by submitting a post; terminated due to LLM API error from content filter.
– Step 82: LLM API error due to content filter. |
| 46 | **Trial 1** (0–42):
– Step 32: current GT; if attributing to this step due to failure to retrieve the desired information, one may attribute earlier steps such as Step 20.
**Trial 2** (43–93): Initially still stuck finding relevant information; later tried a **new** strategy of contacting a potential source via email but was unable to send it.
**Trial 3** (94–123): Continued exploring direct-contact strategies; tried a **new** strategy of live chat and phone call but did not succeed.
**Trial 4** (124–129): Continued direct-contact strategy; terminated due to max rounds reached.
– Step 129: max rounds reached. |
| 47 | **Trial 1** (0–50):
– Step 24: potential error; FileSurfer did not follow the instruction to unzip a file.
**Trial 2** (51–66): Tried a **new** strategy using ComputerTerminal to run code generated by Orchestrator; got a wrong answer due to incorrect code.
– Step 51: current GT; potential error due to wrong code. |
| 51 | **Trial 1** (0–31):
– Step 5: current GT; potential error; FileSurfer unable to transcribe an MP3 audio file.
**Trial 2** (32–46): Tried another device but was not successful.
**Trial 3** (47–99): Attempted to use a **new** strategy of using online service to transcribe audio; not successful.
**Trial 4** (100–122): Continued exploring the online-service approach. |
| 56 | **Trial 1** (0–33):
– Step 4: current GT; no obvious error.
**Trial 2** (34–67): Tried a **different** website; stuck locating the desired information.
**Trial 3** (68–94): Tried **another** website but still could not locate the desired information.
**Trial 4** (95–128): Returned to the initially explored website; deeply explored its content; terminated due to max rounds reached.
– Step 128: max rounds reached. |
| 58 | **Trial 1** (0–22):
– Step 22: current GT; potential error due to unnecessary triggering of replanning.
**Trial 2** (23–81): Repeated steps from Trial 1; **made progress** and explored **new** ways to find the "Regression" label but each attempt failed.
**Trial 3** (82–105): Continued web search for the desired page but landed on the wrong page, leading to an incorrect final result. |
| 7 | GT appears correct. |
| 12 | GT appears correct. |
| 14 | GT appears correct. |
| 24 | GT appears correct. |

*Continued on next page*

*Continued from previous page*

| WW ID | Annotation Notes |
|-------|------------------|
| 26 | Step 32: current GT; LLM API error due to content filter. |
| 29 | Step 12: current GT; LLM API error due to content filter. |
| 33 | Step 8: current GT; LLM API error due to content filter. |
| 34 | Step 4: current GT; LLM API error due to content filter. |
| 37 | **Trial 1** (0–24):
– Step 4: current GT; potential error due to unspecified search query.
**Trial 2** (24–58): Continued the same strategy as Trial 1.
– Step 58: LLM API error due to content filter. |
| 42 | GT appears correct. |
| 43 | GT appears correct. |
| 45 | Step 20: current GT; LLM API error due to content filter. |
| 49 | GT appears correct. |
| 53 | GT appears correct. |
| 54 | GT appears correct. |

## B    PROMPTS USED IN DOVER

In our debugging pipeline, we adopt a layered prompt template design to enable systematic diagnosis and intervention. The *Trial Segmenter* first partitions the full session log into trials according to the "plan–execution" structure, identifying the indices of the initial planning step and each major plan update. It outputs only the step indices and criteria for initial planning and update planning, providing anchor points for subsequent trial-level analysis.

Next, the *Failure Proposer* summarizes each trial by extracting its plan and execution trajectory, reporting success or failure. In case of failure, it locates the earliest error step, identifies the responsible agent, and articulates the error reason—information that drives the subsequent generation of interventions.

The *Intervention Recommender* then generates the minimal executable fix under the combined constraints of task description, ground-truth answer, and localized failure context (previous two steps + failure step). Its output is unified in JSON format, specifying both the intervention category and the proposed replacement text.

In parallel, the *Ground-Truth Milestone Extractor* abstracts each problem and its answer into at most five tool-agnostic milestones, each containing *order*, *title*, *action*, and *result*, to serve as a process-oriented progress standard. The *Milestone Evaluator* then aligns the real execution trace with these milestones, classifying each as *achieved*, *partial*, or *missed*, and detecting whether a "new path" was explored along with its feasibility and contribution—thus providing a quantitative measure of whether substantial progress was made.

Finally, the module *mislocalization or insufficient fix proposer* is applied after re-running the system with the intervention, distinguishing between "success after intervention," "instruction not executed," and "applied but mislocalized/insufficient fix," and providing alignment evidence to support hypothesis validation and close-loop feedback.

## C    ENABLING DOVER ON AUTOGEN2

To assess whether DoVer can be applied beyond Magentic-One, we integrated it with a MathChat multi-agent system built on the AG2 framework for the GSMPlus experiments. This appendix summarizes how we added checkpoint and re-execution functions to AG2 and how DoVer reuses this mechanism with minimal changes.

**Overview of MathChat in AG2.**    MathChat in AG2 follows a group-chat pattern rather than an explicit planner–executor loop as in M1. A manager agent coordinates several specialized workers (e.g., a problem solver, a code executor, and a verifier). At each turn, the manager selects the

| WW ID | Is Current GT Uncertain? | Does GT Step Look Correct? | Can Multi-Failures be Extracted from Multi-Trials? | Has Ambiguity in Agent/Step Attribution? | Is API Error or Flaky Issue Observed? | Trial Count | Ground-truth Mistake Step | GPT-4o-20241120 | | | GPT-5-chat-20250807 | | |
|---|---|---|---|---|---|---|---|---|---|---|---|---|---|
| | | | | | | | | Run 1 | Run 2 | Run 3 | Run 1 | Run 2 | Run 3 |
| 3 | 1 | 0 | 1 | 1 | 1 | 4 | 32 | 4 | 14 | 14 | 39 | 39 | 39 |
| 5 | 1 | 0 | 0 | 1 | 0 | 1 | 12 | 16 | 16 | 16 | 16 | 16 | 16 |
| 9 | 1 | 0 | 1 | 0 | 0 | 4 | 25 | 42 | 3 | 42 | 26 | 26 | 26 |
| 11 | 1 | 1 | 1 | 0 | 0 | 4 | 24 | 4 | 4 | 4 | 128 | 128 | 128 |
| 20 | 1 | 1 | 0 | 1 | 1 | 2 | 3 | 4 | 4 | 3 | 20 | 20 | 20 |
| 21 | 1 | 1 | 0 | 0 | 1 | 1 | 4 | 4 | 4 | 16 | 24 | 24 | 24 |
| 22 | 1 | 1 | 0 | 1 | 1 | 2 | 4 | 4 | 9 | 20 | 23 | 19 | 23 |
| 27 | 1 | 1 | 0 | 1 | 1 | 2 | 4 | 4 | 16 | 24 | 20 | 20 | 20 |
| 41 | 1 | 1 | 1 | 0 | 1 | 4 | 8 | 16 | 6 | 5 | 4 | 4 | 4 |
| 46 | 1 | 0 | 1 | 0 | 0 | 2 | 32 | 4 | 58 | 63 | 129 | 129 | 129 |
| 47 | 1 | 0 | 1 | 0 | 0 | 4 | 51 | 16 | 4 | 4 | 59 | 59 | 59 |
| 51 | 1 | 0 | 1 | 0 | 0 | 4 | 5 | 5 | 5 | 5 | 5 | 5 | 5 |
| 56 | 1 | 1 | 1 | 0 | 0 | 4 | 4 | 4 | 4 | 11 | 11 | 11 | 11 |
| 58 | 1 | 0 | 1 | 0 | 0 | 3 | 22 | 4 | 4 | 4 | 39 | 39 | 39 |
| 7 | 0 | 0 | 0 | 0 | 0 | 1 | 8 | 4 | 4 | 4 | 4 | 4 | 4 |
| 12 | 0 | 0 | 0 | 0 | 0 | 1 | 16 | 16 | 16 | 16 | 16 | 16 | 16 |
| 14 | 0 | 0 | 0 | 0 | 0 | 1 | 14 | 31 | 31 | 30 | 14 | 14 | 14 |
| 24 | 0 | 0 | 0 | 0 | 0 | 1 | 1 | 1 | 1 | 1 | 1 | 1 | 1 |
| 26 | 0 | 0 | 0 | 0 | 1 | 1 | 32 | 16 | 16 | 16 | 16 | 16 | 16 |
| 29 | 0 | 0 | 0 | 0 | 1 | 1 | 12 | 4 | 4 | 4 | 6 | 6 | 10 |
| 33 | 0 | 0 | 0 | 0 | 1 | 1 | 8 | 4 | 4 | 4 | 4 | 4 | 4 |
| 34 | 0 | 0 | 0 | 0 | 1 | 1 | 4 | 4 | 2 | 4 | 6 | 0 | 0 |
| 37 | 0 | 0 | 0 | 0 | 1 | 2 | 4 | 4 | 4 | 3 | 4 | 4 | 4 |
| 42 | 0 | 0 | 0 | 0 | 0 | 1 | 29 | 29 | 28 | 4 | 29 | 29 | 29 |
| 43 | 0 | 0 | 0 | 0 | 0 | 1 | 12 | 12 | 12 | 12 | 12 | 12 | 12 |
| 45 | 0 | 0 | 0 | 0 | 1 | 1 | 20 | 4 | 6 | 8 | 1 | 1 | 1 |
| 49 | 0 | 0 | 0 | 0 | 0 | 1 | 12 | 12 | 12 | 12 | 12 | 12 | 12 |
| 53 | 0 | 0 | 0 | 0 | 0 | 1 | 24 | 24 | 24 | 24 | 24 | 24 | 24 |
| 54 | 0 | 0 | 0 | 0 | 0 | 1 | 15 | 16 | 16 | 16 | 16 | 16 | 16 |

Table 7: Tagging results for each GAIA case in WW. The table reports ground-truth annotations, presence of ambiguous attributions, uncertainty tags, possibility for multi-failure step attribution, potential API or flaky errors, and case-specific details such as number of trials and model outputs. Model predictions matching the ground-truth failure step are highlighted in green.

next speaker based on the conversation state. There is no dedicated "planner" agent; instead, new reasoning attempts or verification cycles emerge when different specialists are invoked.

**Enhancing AG2 with Checkpointing and Replay.** Since AG2 does not natively support checkpointing and replay functions, we implemented a lightweight checkpointing layer around the AG2 conversation manager. Specifically, after each agent turn, we serialize the full logical state needed to resume the run, including:

- the conversation history up to that step (messages, speaker identities, and any tool outputs);

- the configuration of all agents in the chat (roles, prompts, tool bindings);

- the underlying LLM configuration (model name, temperature, decoding parameters).

For MathChat, there is no persistent external environment beyond these components, so we do not need to snapshot additional tool state. Each checkpoint is stored as a structured object keyed by the step index, enabling us to load the system state at any past turn. We have released this AG2 enhancement with checkpointing and replay functions in our anonymous repository linked in this paper.

**Enabling DoVer on MathChat in AG2.** To run DoVer interventions, we load the checkpoint corresponding to the target step, reconstruct the manager and agents from the serialized state, and then apply the intervention by editing the message at the target step (for example, replacing the manager's instruction with the text proposed by DoVer). We then resume the conversation from the target step onward using the original AG2 run loop. Because interventions operate only at the message-passing layer over restored state, this integration does not require changes to AG2's core scheduling or tool interfaces; the DoVer pipeline, including trial segmentation and intervention generation, can be reused unchanged after adapting prompts to the MathChat setting.

**Implementation Effort.** The AG2 integration required adding a checkpoint-aware wrapper around the conversation manager and a small amount of glue code to bridge checkpoints with DoVer's intervention executor. In practice, this amounted to on the order of one thousand lines of code and a few days of work by a single engineer, aided by LLM-based coding assistance. Once this infrastructure is in place, plugging DoVer into a new AG2-based multi-agent application mainly requires registering checkpoint capture, exposing a replay entry point from a given step index, and mapping DoVer's intervention categories to concrete message edits at that step.

**Web-based Intervention Interface for AutoGen2** On top of this checkpoint and replay infrastructure, we implemented a minimal web-based user interface for the AG2 MathChat system (Figure 4). The dashboard exposes the same DoVer functionality used in our experiments: the left panel lists past sessions and checkpoints, the top bar accepts a new math problem, the central pane visualizes the multi-agent dialogue, the right "Intervention" pane lets a user select a step, edit the corresponding message or plan, and resume execution from that point, and the bottom area records continuation histories after each intervention. This interface is intended purely as a lightweight visualization and human-in-the-loop debugging tool; no experiment logic depends on it.

**Guidelines for Enabling DoVer in Other Agent Frameworks.** We highlight several design principles from the above integration experience:

1. *Minimal-invasive Design*: By wrapping core components (e.g., chat manager in AG2) with checkpoint-aware versions, we preserved existing architecture while enabling checkpointing.

2. *Capture Full System State*: We used LLM-assisted tooling to identify all relevant state variables (e.g., agent configs, LLM settings, conversation history), ensuring accurate restoration.

3. *Intervention via Checkpoint Manipulation*: Interventions operate directly on structured checkpoint data, enabling seamless reuse of restoration and replay logic.

We hope these guidelines make it easier to integrate DoVer into a wide range of agent systems.

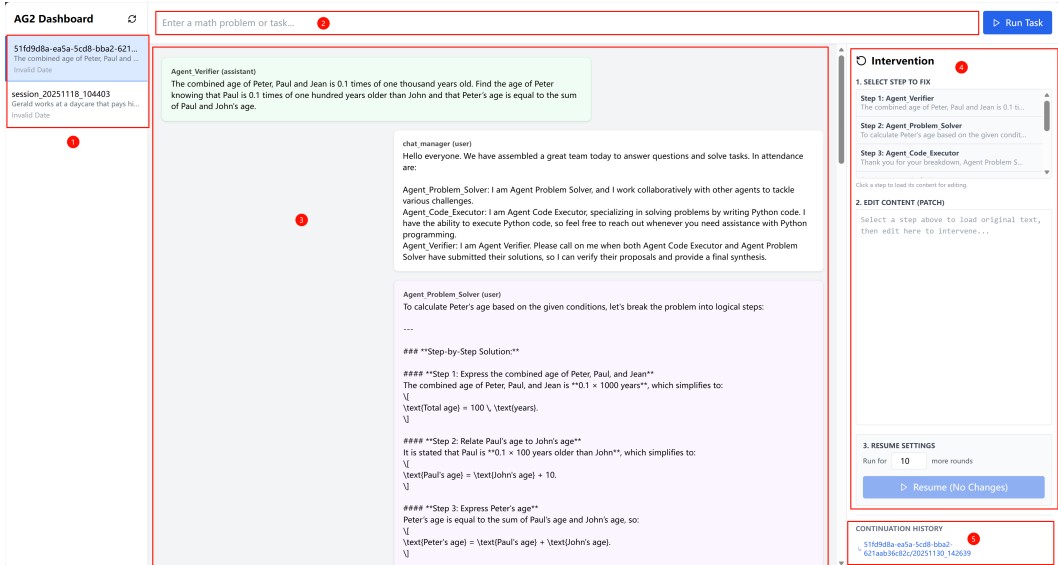

Figure 4: Web-based intervention user interface for the AG2 MathChat system. (1) List of recorded math problem sessions. (2) Input box for submitting a new math task. (3) Main conversation panel showing the multi-agent trace and intermediate reasoning. (4) Intervention panel, where a user can select a specific agent and step to edit the message or plan. (5) History panel showing checkpoints and continuations after interventions.

## D    QUALITATIVE CASE STUDIES

We present four representative case studies of the above intervention outcome category. Each vignette briefly introduce the task context, the hypothesized failure point, the concrete intervention applied, and the observed outcome.

### (1) Refuted: Case 46 (Trial 1) in WW-GAIA

- **Task:** What time was the Tri-Rail train that carried the most passengers on May 27, 2019, scheduled to arrive in Pompano Beach?

- **Diagnosis:** The failure proposer hypothesized that the issue arose from *WebSurfer*'s inability to locate a data file at a particular step, even though *WebSurfer* had in fact already opened that file.

- **Intervention:** *WebSurfer* was explicitly instructed to open the file that it had already accessed.

- **Effect:** Although the intervention generator produced instructions consistent with the hypothesized failure, and the agents followed these instructions faithfully, the trials still failed. Because the agent correctly executed the intervention and the intervention accurately reflected the failure hypothesis, the resulting failure demonstrates that the original hypothesis was invalid. We therefore classify this trial as *refuted*.

### (2) Inconclusive: Case 21 (Trial 2) in WW-GAIA

- **Task:** Locate the paper linked at the bottom of Carolyn Collins Petersen's June 6, 2023 *Universe Today* article and identify the NASA award number that supported R. G. Arendt.

- **Diagnosis:** The *Orchestrator* remained stuck in the planning phase and never activated *WebSurfer* or other agents. No evidence was collected, leaving the trial incomplete.

- **Intervention:** The *WebSurfer* was told to bypass incremental scrolling and jump straight to the article's footer to scan for DOIs, arXiv links, or other research references.

- **Effect:** The *WebSurfer* could not execute the "scroll-to-bottom" action. Instead, it performed a single page scroll without reaching the references, so the plan failed to advance. The inconclusive

outcome reflected tool constraints rather than wrong failure hypothesis or intervention. Correct strategy was blocked by limited execution abilities, pointing to the need for stronger action primitives.

Across these four cases, **DoVer** exhibits all intended outcomes: (i) *validated* when a targeted edit repairs the trajectory; (ii) *partially validated* when the edit induces measurable progress yet external frictions block completion; (iii) *refuted* when faithful execution of the edit leaves the failure state intact; and (iv) *inconclusive* when tooling prevents faithful execution of the intended intervention.

### (3) Validated: Case 3 (Trial 1) in WW-GAIA

**- Task:** Identify the architectural firm associated with a Chicago landmark referred to indirectly via a NASA APOD entry from Aug. 1–7, 2015, as shown in Figure 1.

**- Diagnosis:** The *WebSurfer* agent engaged in prolonged, aimless scrolling of the APOD archive and repeatedly missed the required date-bounded entries, despite receiving explicit guidance.

**- Intervention:** We replaced unstructured scrolling with an in-archive, date-bounded keyword strategy: *"Search the APOD archive directly, restrict to August 1–7, 2015, and scan for 'city lights' / 'horizon'; report the entry title, date, and a brief summary"*.

**- Effect:** The system revised its plan to prioritize date/keyword filters, issued a targeted query, and quickly converged along the reasoning chain *Marquette (city)* $\rightarrow$ *Jacques Marquette (namesake)* $\rightarrow$ *Marquette Building (Chicago)* $\rightarrow$ *Holabird & Roche (architects)*, yielding the correct first name "Holabird." The failed trial flipped to success, *validating* both the hypothesis (strategy error) and the remedy (focused retrieval).

### (4) Partially Validated: Case 56 (Trial 3) in WW-GAIA

**- Task:** According to Google Finance, when was the first year the Apple stock went above $50 (without adjusting for stock split)?

**- Diagnosis:** Execution stalled while probing Alpha Vantage as a source for unadjusted historical prices; the agent failed to complete source verification.

**- Intervention:** We redirected exploration *away* from generic search results and *toward* Alpha Vantage's homepage, instructing the agent to verify data availability and note access requirements.

**- Effect:** The trajectory moved onto the correct information source, but repeated script errors and API key frictions blocked completion. The outcome was clear forward progress without a final result and the hypothesis was partially validated since the re-planning was correct but execution was constrained by environment and tooling.

## E  ANALYSIS OF INCONCLUSIVE CASES AND TARGETED TOOL ENHANCEMENT

The 29–67% *Inconclusive* cases in Table 3 reveal the limits of orchestrator-level interventions: many failures arise from sub-agent capability gaps the orchestrator cannot resolve. These cases highlight the boundaries of sub-agent competence and indicate the improvement directions. We thus view DoVer as part of a larger debugging loop: (i) DoVer provides automated orchestrator-level interventions; (ii) unresolved failures surface sub-agent weaknesses requiring human or auto refinement.

To illustrate the effectiveness of this "DoVer automation + human-in-the-loop" cycle, we analyzed the failure traces of inconclusive cases in WW-GAIA and discovered two recurring failure modes in *WebSurfer*: (i) missing a "scroll-to-bottom" tool, causing repeated partial scrolling, and (ii) inability to process PDFs, leading to empty summaries. We implemented targeted fixes by adding the scrolling tool and revising PDF handling. After these upgrades, previously inconclusive Cases 20, 21, and 26 can now be solved using only DoVer's orchestrator-level interventions. This shows that DoVer not only repairs recoverable failures but also surfaces concrete sub-agent bottlenecks.

We see two future work directions depending on the degree of permissible system modification: (i) fully automated debugging loops, where DoVer's insights feed into automated sub-agent improvement (e.g., using a coding agent to implement fixes). (ii) capability-aware intervention generation, allowing DoVer to tailor interventions to known sub-agent limits when code changes are not allowed. Both directions support the goal of building robust, self-improving multi-agent systems.

```
                                    Trial Segmenter Prompt

- You are an experienced system log analyzer and decomposer, who can effectively decompose and organize complex session
logs into meaningful components.
- Your task is to analyze a session log from a Large Language Model (LLM) based agent system, and decompose the session
log into a series of trials that the system had made to solve a given task.
- Refer to Section `On the LLM-based Agent System` for details about the LLM-based agent system.
- Refer to Section `On the Input Data Structure` for details on the data structure of the input logs.
- Refer to Section `Guidance for Log Decomposition` for details on how to perform the log decomposition.
- Refer to Section `On the Output Format` for details on the output data structure.

# On the LLM-based Agent System
- The LLM-based agent system can be a single or multi-agent system.
- The system is often tasked with a problem, and agents in the system work collectively to solve the task.
- Agents in the system can have different roles and responsibilities, and they may need to communicate and collaborate
with others to achieve the overall goal.
- Agents work in the SEQUENTIAL manner, namely, agents are invoked one by one and no agents are executing in parallel.
- There often exists an agent playing the role of Orchestrator, who devises and updates a plan for solving the task and
coordinates with the agents to solve the task.
- The general flow of the system follows the cycle of "make a plan", "execute plan", "update plan", and so on.

# On the Input Data Structure
- The input data are collected from a session in the LLM-based agent system.
- The session contains a series of interactions among agents in the system. The interaction often starts with a specific
task and the goal of the system is to finish task successfully.
- The input data are in the following Json format:
{
    'problem': <problem>,
    'ground_truth_answer': <ground_truth_answer>,
    'session_logs': <session_logs>
}
- `problem` denotes the specific task that the agent system needs to solve.
- `ground_truth_answer` is the corresponding ground truth answer for the problem.
- `session_logs` provides the details of the session trace, and is a string object with the following format:
'''
    [Step 0] "agent a": "xx",
    [Step 1] "agent b": "xxxx",
    [Step 2] "agent c": "xxxxx",
    [Step 3] "agent a": "xxxxxxx"
'''
Each entry in the above represents a `Step` where an agent provides input. The 'x' symbolizes the speech/message of each
agent.

# Guidance for Log Decomposition
- The goal is to decompose the session logs into a series of `Trial`s that had been made by the system.
- Each `Trial` is defined as an attempt to solve the problem, consisting of a `Planning` step and its corresponding
`Execution` steps.
- Both `Planning` and its `Execution` are manifested by `Step`s in the log.
- The `Planning` steps outline the strategy for solving the problem, while the `Execution` steps are the actual attempts
made by the agents to implement the plan.
- `Planning` steps include both the initial planning and any subsequent updates to the plan.
- If `Planning` is updated in the log (e.g., due to no progress made after executing current plan), NEW `Trial` should be
considered.
- The key of log decomposition is to categorize all `Step`s in the log into Four categories, `Initial_Planning`,
`Update_Planning`, `Execution`, and `Others`. Refer to the following guidance for the categorization:
- `Initial_Planning` step are the step taken to create a plan for solving the problem. These steps typically involve
brainstorming, outlining step-by-step strategies, and setting goals. Only ONE step in the log should be classified as
`Initial_Planning`.
- `Update_Planning` steps are those that modify or refine the existing plan based on new information or feedback. These
steps may involve re-evaluating the strategy, adjusting goals, or incorporating lessons learned from previous attempts.
Note that the `Update_Planning` step refers to the major update of the whole task plan. The small changes of in the
execution of a plan (e.g., clicking another links in the search result page when the previous link click returns no useful
result) do NOT count as the `Update_Planning` step, as it is still part of a plan execution but just a different execution
detail.
- Both `Initial_Planning` and `Update_Planning` steps are often carried out by the agent of the Orchestrator role. But not
all `Step`s from the Orchestrator agent are `Planning` steps, as the Orchestrator agent can have responsibilities other
than planning. For example, when executing a plan, the Orchestrator agent may provide instructions and guidance to other
agents for better plan execution or coordination among agents.
- Not every `Initial_Planning` or `Update_Planning` step is required to be followed by `Execution` steps. This can occur
if the plan is not executed or if the execution details are not captured in the log. Thus, it is acceptable to have
several consecutive steps are labelled as `Planning` steps.
- `Others` steps are auxiliary steps that do not directly contribute to the planning or execution of the task. These may
include acknowledgments of the task receival, final result collection and report result to the user, or other non-
essential messages related to the core task-solving process. They often appear at the begining and end of the session
logs.
- Follow the below steps to decompose the session logs into `Trial`s:
- 1. Read through the `session_logs` to understand the overall context. Specifically,
    - Understand the task and identify the set of agents has been assembled to solve the task.
    - Identify the specific roles and responsibilities of each agent in the context of the task. Identify the Orchestrator
agent and how it coordinate other agents.
- 2. Categorize each `Step` in the log into `Initial_Planning`, `Update_Planning`, `Execution` and `Others` categories
based on the guidance provided above.
- 3. Only output the step indices of the `Initial_Planning` and `Update_Planning` steps and the reasons for their
categorization.

# On the Output Format
- The output should be a Json object with the following structure:
{
    'initial_planning_step': {
    'step_index': <the original step index of the initial planning step, e.g., 0>,
    'reason': <the reason why this Step 0 is identified as an initial planning step>
    },
    'plan_update_steps': [
    ...,
    {
        'plan_update_step_index': <the original step index of the plan update step, e.g., 5>,
        'reason': <the reason why this Step 5 is identified as a plan update step>
    },
    ...
    ]
}
```

Figure 5: Trial segmenter prompt: log decomposition of full session into planning–execution trials.

---

**Failure Proposer Prompt 1**

- You are an experienced system log analyzer and summarizer, who can effectively analyze and summarize complex session logs into concise insights.
- Your task is to analyze a partial session log related to a trial to solve a given task in a Large Language Model (LLM) based agent system, and summarize the key process of the trial as well as perform root cause analysis of the trial if it is a failed trial.
- Refer to Section `On the LLM-based Agent System` for the background knowledge about the LLM-based agent system.
- Refer to Section `On the Trial and Session Log Decomposition` for the background knowledge about log of a session and how the prior process of decomposing the whole session log into different trials.
- Refer to Section `On the Input Data Structure` for details on the input data for the current trial summarization task.
- Refer to Section `Guidance for Trial Summarization` for details on how to perform the log analysis and trial summarization.
- Refer to Section `On the Output Format` for details on the output data structure.

# On the LLM-based Agent System
- The LLM-based agent system can be a single or multi-agent system.
- The system is often tasked with a problem, and agents in the system work collectively to solve the task.
- Agents in the system can have different roles and responsibilities, and they may need to communicate and collaborate with others to achieve the overall goal.
- Agents work in the SEQUENTIAL manner, namely, agents are invoked one by one and no agents are executing in parallel.
- There often exists an agent playing the role of Orchestrator, who devises and updates a plan for solving the task and coordinates with the agents to solve the task.
- The general flow of the system follows the cycle of "make a plan", "execute plan", "update plan", and so on.

# On the Trial and Session Log Decomposition
- The input trial log data are part of a session in the LLM-based agent system.
- The session contains a series of interactions among agents in the system. The interaction often starts with a specific task and the goal of the system is to finish task successfully.
- The whole log of the session (`session_logs`) is structured as the step-by-step messages from agents, namely,
```
[Step 0] "agent a": "xx",
[Step 1] "agent b": "xxxx",
[Step 2] "agent c": "xxxxx",
[Step 3] "agent a": "xxxxxxx"
```
Each entry in the above represents a `Step` where an agent provides input. The 'x' symbolizes the speech/message of each agent.
- The whole session log had been decomposed into different `Trial`s with the following requirements:
- Each `Trial` is defined as an attempt to solve the problem, consisting of a `Planning` step and its corresponding `Execution` steps.
- Both `Planning` and its `Execution` are manifested by `Step`s in the log.
- The `Planning` steps outline the strategy for solving the problem, while the `Execution` steps are the actual attempts made by the agents to implement the plan.
- `Planning` steps include both the initial planning and any subsequent updates to the plan.
- Going through the session log, if there is a step related to plan update (e.g., due to no progress made after executing the existing plan), the existing `Trial` would terminate and a new `Trial` would be considered to start.
- The decomposition of the session log into `Trial`s allows for a more granular analysis of the agent system's behavior and performance.

# On the Input Data Structure
- The input data are in the following Json format:
```
{
    'problem': <problem>,
    'ground_truth_answer': <ground_truth_answer>,
    'previous_trial_summary': [
    ...,
    {
        'trial_index': <trial_index>,
        'trial_plan': <trial_plan>,
        'trial_execution': <trial_execution>,
        'is_succeed': <is_succeed>,
        'trial_summary': <trial_summary>
    },
    ...
    ],
    'trial_logs_to_summarize': <trial_logs_to_summarize>,
    }
}
```
- `problem` denotes the specific task that the agent system needs to solve.
- `ground_truth_answer` is the corresponding ground truth answer for the problem.
- `previous_trial_summary` provides the history of all previous trials in the session. Each element corresponds to each historical trial, including their `trial_index`, `trial_plan` (i.e., step by step plan for the trial), `trial_execution` (i.e., execution details for the trial), `trial_summary` and the indicator of 'is_succeed' to denote whether the trial successfully solves the task or not. Empty list for the first trial.
- `trial_logs_to_summarize` provides the log details related to the trial intended to be summarized. It still follows the above step-by-step style of `session_logs` but only contains the steps related to the specific trial.
```
    [Step n] "agent a": "xx",
    [Step n+1] "agent b": "xxxx",
    ...
    [Step m] "agent c": "xxxxxxx"
```
The initial step of the trial, (i.e., `Step n`) often refers to the above `Planning` step, involving an initial plan creation or update, and the following steps belong to the execution of the plan of this trial.

Figure 6: Trial summarizer + failure proposer prompt: per-trial summary and root-cause localization, part 1.

---

**Failure Proposer Prompt 2**

# Guidance for Trial Summarization
- The goal is to summarize the provided `trial_logs_to_summarize` based on the overall context including the task and previous trial history.
- Follow the below steps for trial summarization:
- 1. Read through the `problem` and `ground_truth_answer` to understand the task and its ground truth answer.
- 2. Read through the `previous_trial_summary` to understand how agents try to solve the task in previous trials.
- 3. Examine the `trial_logs_to_summarize` to understand the specific steps taken in the current trial. Specifically,
    - Understand the first planning step of the current trial. If it involves a plan update, you need to derive the reasoning behind the update by considering the `trial_plan` and `trial_summary` of the last trial.
    - Closely track the execution steps after the planning step of the current trial and examine how they carry out the plan and solve the task. Note that there can be NO execution steps after the planning step, e.g., due to no execution in reality.
    - Pay high attention to the final outcome of the trial, especially whether it successfully solve the task or not. If succeed, please reflect how and why this trial solves the task successfully, and compare its success with the prior failed trials. If not, please reflect why it fails and which mistake agents and steps are responsible for the failure.
- 4. Output the trial summary with the following key components:
    - `trial_context`: the context in which the trial was conducted, including relevant information from previous trials and the current task.
    - `trial_plan`: a detailed plan for the current trial. The plan should be self-contained, namely, if the original log at the plan step only contains the plan update part, you should derive the missing context from the previous trial's plan and summary. Output the plan in the step-by-step format and number them for the progress tracking, e.g., ["1. xxx", "2. xxx", ...].
    - `trial_execution`: a detailed description of the execution process during the CURRENT trial, NOT the whole session. Only extract execution details from the logs given in the current trial logs (`trial_logs_to_summarize`). NEVER quote execution details from prior trials. If no execution details can be found after the planning step in the current trial (e.g., due to no execution in reality), directly output "No execution details found".
    - `plan_fulfillment_status`: summarize the plan fulfillment status based on `trial_execution` and `trial_plan`. You need to map the progress from execution to the numbered planned steps in `trial_plan`. Namely, output the plan fulfillment status as a json object:
        {
            "fully_fulfilled_plan_steps": [<plan_step_index>],
            "partially_fulfilled_plan_steps": [<plan_step_index>],
            "unfulfilled_plan_steps": [<plan_step_index>]
        }
    Note that `<plan_step_index>` refers to the plan step index in `trial_plan`, not the step index in the original session logs.
    - `is_succeed`: whether the current trial successfully solves the task or not.
    - `trial_reflection`: a reflection on the trial's outcomes, including what worked well, what didn't, and any insights gained for future trials.
    - `mistake_agent`: only required for failed trials. Identify which agent is mainly responsible for the trial failure. If multiple agents are responsible, only choose the most relevant one.
    - `mistake_step_index`: only required for failed trials. Identify the specific step in the trial where the mistake occurred.
    - `mistake_reason`: only required for failed trials. Explain why you choose the mistake agent and step.
    - 'trial_overall_summary': a summary of the trial's overall performance, including key process, successes and failures.

# On the Output Format
- The output should be a Json object with the following structure:
{
    'trial_context': <the context in which the trial was conducted. Object Type: string>,
    'trial_plan': <a detailed plan for the current trial. Object Type: list of string>,
    'trial_execution': <a detailed description of the execution process and progress made during the trial. Output "No execution details found" if no execution steps after the planning step. Object Type: string>,
    'trial_progress': {
    "fully_fulfilled_plan_steps": [<plan_step_index, Object Type: int>],
    "partially_fulfilled_plan_steps": [<plan_step_index, Object Type: int>],
    "unfulfilled_plan_steps": [<plan_step_index, Object Type: int>]
    },
    'is_succeed': <whether the current trial successfully solves the task or not. Object Type: boolean>,
    'trial_reflection': <a reflection on the trial's outcomes. Object Type: string>,
    'mistake_agent': <the agent mainly responsible for the trial failure, if applicable. Object Type: string>,
    'mistake_step_index': <the specific step in the trial where the mistake occurred, if applicable. Object Type: int>,
    'mistake_reason': <an explanation of why the identified agent and step are considered mistakes, if applicable. Object Type: string>,
    'trial_overall_summary': <a summary of the trial's overall performance, including key process, successes and failures. Object Type: string>
}

Figure 7: Trial summarizer + failure proposer prompt: continuation of Figure 6, part 2.

---

**Intervention Recommender Prompt**

```
You are an expert in debugging multi-agent systems. A failure proposer has analyzed a Magentic-One execution and identified a
problematic step.
Use this as guidance to design a precise, minimal intervention that fixes the root cause.

## Task Context
Here is the original task: {problem}
And here is the ground truth for guidance: {ground truth}
## Failure Analysis
Failed step index: {step index}
Failed agent: {agent}
Diagnosed reason: {reason}
## Execution Context
Here are the two steps immediately before the failure and the failed step itself:
{Two steps context}

## System specifics and intervention policy:
- If failure is in Orchestrator: Task Full Ledger issues -> provide ONLY the minimal replacement snippet(s) for the affected
section(s): Facts and/or Plan. Use tags [FACTS_REPLACEMENT]: ... and/or [PLAN_REPLACEMENT]: ...; do not output the entire Task
Full Ledger. Use this only when the failed content is the Task Full Ledger content (starts with 'We are working to address the
following user request:' and includes the Facts/Plan sections).
- If Orchestrator instruction is wrong/ambiguous -> provide the exact corrected instruction as a single atomic next step. If the
message is not the Task Full Ledger scaffold, or when unsure, treat it as orchestrator_instruction.
- If a subagent failed -> infer, from the provided context, how that subagent or tool should have acted; rewrite the
Orchestrator's instruction to that subagent/tool so that it leads to the correct behavior.
- Keep changes minimal and targeted. Avoid global resets.
- Do not give any ground truth in the intervention message.
Return STRICT JSON with the following schema:
{
  "category": one of ["orchestrator_ledger", "orchestrator_instruction", "subagent_instruction"],
  "replacement_text": "exact text that should replace the problematic content"
}
Respond with JSON only, no extra commentary.
```

Figure 8: Intervention recommender prompt: minimal, executable fix classification; JSON category and replacement text.

```
                                    Milestone extractor Prompt

## Role & Task Introduction
You are **Milestone Extractor**, a precise formatter that converts a research QA task into a compact set of high-level
milestones. Given a **question**, its **ground-truth answer**, and **human annotation steps** (which may include tool
usage), produce a short, ordered list of the *key individual milestones* needed to achieve the answer. Focus on
abstraction and outcome, not tool-specific clicks.

Your goal: **summarize the path to the answer in ≤5 milestones** that capture the essential reasoning and actions,
suitable for audit or scripting.

## Inputs

You will receive a single JSON object with the following fields:

- `question` *(string, required)* — The task question.
- `ground_truth_answer` *(string, required)* — The validated answer.
- `human_annotation_steps` *(string or object, optional)* — Notes describing how a human solved it (may include step
lists, links, or tool mentions).

### Input Assumptions
- `human_annotation_steps` may be verbose or noisy; assume it's trustworthy but not necessarily well-structured.
- If `human_annotation_steps` is missing or incomplete, infer reasonable milestones from the `question` and the nature of
the task.

### Example Input Format
**Minimal (string steps):**
```json
{
  "question": "…",
  "ground_truth_answer": "…",
  "human_annotation_steps": "Searched X → Identified Y → Opened Z → Extracted answer."
}
```

## Guidance

1. **Abstract, don't recount.**
   Convert granular browsing/clicking into conceptual milestones (e.g., "Resolve riddle of journal identity" rather than
"Open Wikipedia page X").

2. **Preserve causal order.**
   Maintain the minimal sequence from problem framing → locating the source → isolating evidence → confirming the answer.

3. **Cap at five milestones.**
   If more than five logical steps exist, merge adjacent ones by theme (e.g., "Locate issue & open target article").

4. **Use consistent fields.**
   Each milestone must include:
   - `order` *(1-based integer)*
   - `title` *(short noun phrase)*
   - `action` *(what to do; imperative, concise)*
   - `result` *(what this step yields toward the answer)*

5. **Be tool-agnostic.**
   Avoid naming specific search engines, sites, clicks, or UI elements unless essential for understanding.

6. **Echo the answer.**
   Ensure the final milestone logically confirms or extracts the `ground_truth_answer`.

7. **Clarity over completeness.**
   Prefer short, crisp phrasing. No extraneous commentary, citations, or links.

## Output Format

Produce **only** a JSON object with the following structure:

```json
{
  "question": "<string>",
  "ground_truth_answer": "<string>",
  "milestones": [
    {
      "order": 1,
      "title": "<short title>",
      "action": "<imperative action>",
      "result": "<concise outcome>"
    }
    // up to 5 total milestones
  ]
}
```
```

Figure 9: Ground-truth milestone extractor prompt: $\leq 5$ tool-agnostic milestones.

```
                              Milestone Evaluator Prompt

## Role & Task Introduction
You are **SessionLog Milestone Evaluator**, a precise formatter that analyzes an LLM-agent session log against a
predefined set of milestones and outputs a compact progress report plus an assessment of any alternate ("new") exploration
paths taken.

Your goal: **produce a single JSON object** containing:
- `milestone_progress`: status for each milestone (≤5 milestones total)
- `new_path_assessment`: a unified evaluation of alternate paths explored in the log

## Inputs
You will receive a single JSON object with the following fields:
- `question` *(string, optional)* — The task question (for context).
- `ground_truth_answer` *(string, optional)* — The validated answer (for context).
- `milestones` *(array, required)* — The authoritative list of milestones to evaluate. Each item includes:
  - `order` *(int, 1-based)*,
  - `title` *(string)*,
  - `action` *(string)*,
  - `result` *(string)*.
  - **Note:** If more than 5 milestones are provided, evaluate only the first five.
- `session_log` *(array, required)* — The actual agent trace. Entries may include fields like `name`, `content`,
`timestamp`, `step_idx` (values may vary).

### Input Assumptions
- The `milestones` list is authoritative; **do not invent, merge, or reorder milestones**.
- The `session_log` may be verbose, noisy, or partially redundant.
- Evidence should be grounded **only** in the provided `session_log` (no external browsing or assumptions).

### Example Input Format
**Minimal (string steps):**
```json
{
  "question": "…",
  "ground_truth_answer": "…",
  "milestones": "[...]",
  "session_log": "[...]"
}
```

## Evaluation Rules

### 1) Milestone Progress
For each milestone:
- Determine `status` as one of:
  - `achieved` — clear evidence the milestone's intended **result** occurred.
  - `partial` — meaningful progress toward the result, but incomplete.
  - `missed` — no sufficient evidence of progress toward the result.
- Provide `evidence`:
  - Be concise (1–2 sentences).
  - Quote short phrases from the log **or** reference `step_idx` values where available (e.g., "step_idx 16–18").
  - Explain *why* the status is assigned (what the log shows or fails to show).

### 2) New Path Assessment
Identify whether the session explored **alternate paths** not implied by the milestones' normal route (e.g., detours to
secondary indexes, different tools, or file-download attempts not required by the milestones).
- `is_new_path_explored` *(boolean)* — true if such deviations exist.
- `evidence` *(string)* — brief proof from the log (e.g., tool/site names, failed downloads, HTTP errors), ideally with
`step_idx` references.
- `is_viable` *(boolean)* — whether the alternate path is **in principle** a reasonable way to solve the task (even if it
failed here).
- `viability_evidence` *(string)* — justification for `is_viable` grounded in the log (e.g., "issue page offered a PDF
link; using full-issue PDFs is a common route").
- `is_successful` *(boolean)* — whether the alternate path **actually yielded** the required progress or answer in this
session.
- `success_evidence` *(string)* — concise support for `is_successful` (e.g., "FileNotFoundError persisted; no quotation
extracted").

### Classification Hints
- Prefer `achieved` only when the milestone's **result** is explicitly met (not just attempted).
- Use `partial` when the agent reached the correct resource or intermediate state but didn't complete the intended
extraction/verification.
- Use `missed` when there's no concrete evidence the agent moved toward the intended outcome.
- A "new path" is about **method divergence**, not minor navigation within the same path.

## Output Format
Produce **only** a JSON object with the following structure (no extra text, comments, or markdown fences):

```json
{
  "milestone_progress": [
    {
      "order": 1,
      "milestone": "<title from milestones[i].title>",
      "status": "achieved | partial | missed",
      "evidence": "<brief, log-grounded justification (may include step_idx refs)>"
    }
    // ... up to 5 items total
  ],
  "new_path_assessment": {
    "is_new_path_explored": true,
    "evidence": "<brief proof from session_log>",
    "is_viable": true,
    "viability_evidence": "<why this method is generally reasonable, grounded in the log>",
    "is_successful": false,
    "success_evidence": "<why it did or did not work in this session>"
  }
}
```

## Style & Constraints
- Be concise and neutral; no speculation beyond the log.
- Do not include URLs, citations, or external sources outside what's quoted from session_log.
- Do not exceed two sentences per evidence field.
- Do not add fields or commentary outside the specified schema.
```

Figure 10: Milestone evaluator prompt: progress vs milestones and new-path assessment.

---

**Mislocalization or Insufficient Fix Proposer Prompt**

```
You are an expert judge analyzing post-intervention runs.

Input is a usr_msg JSON object with keys: problem, ground_truth_answer, initial_session_log, initial_failure_analysis,
intervention_details, post_intervention_runs (a single run: list with one {session_log}).

Use all content as-is without truncation.

Classify the single run and the overall outcome into exactly one of:

- success_after_intervention: the subagent explicitly follows the new intervention instruction and the final outcome is correct.

- execution_issue_intervention_not_applied: the subagent does not follow the new intervention instruction; the outcome remains
incorrect.

- proposer_mislocalization_or_insufficient_fix: the intervention is observed/applied but the injection point/fix is wrong; the
outcome remains incorrect.

For each run, write a concise reason that includes these parts: before_intervention: <what the system/subagent did that led to
failure>, after_intervention: <what changed in this run after the injection>, system_action: <whether/how the subagent followed
the new instruction>, final_outcome: <correct/incorrect vs ground_truth>.

Strictly output JSON:
{
  "overall":
    {
        "label","reason","evidence":{"quotes_or_tokens":[],"ground_truth_match":bool,"intervention_applied":bool}
    }
}.

Evidence should include 1-3 verbatim quotes from the logs.
```

Figure 11: Post-intervention outcome classifier prompt: mislocalization or insufficient fix proposer.

