# OpenReview forum: "DoVer: Intervention-Driven Auto Debugging for LLM Multi-Agent Systems"
_ICLR.cc/2026/Conference — ICLR 2026 Poster_

### Official Review · Reviewer_tv3u · 2025-10-30

**Soundness:** 3
**Presentation:** 3
**Contribution:** 3
**Rating:** 6
**Confidence:** 5

**Summary:**

This paper introduces DoVer, a novel framework for debugging failures in LLM-based multi-agent systems through a process of intervention-driven hypothesis validation. Instead of relying solely on log-based failure attribution, DoVer segments execution logs into trials, generates candidate failure hypotheses, applies targeted interventions, and then re-executes the system to assess if the intervention leads to task recovery or meaningful progress. Through extensive experiments on failure datasets derived from GAIA and AssistantBench, the authors show that DoVer can flip 18–28% of failures into successes and validate or refute 30–60% of failure hypotheses.

**Strengths:**

1. The authors rigorously analyze the shortcomings of current log-based debugging benchmarks and reveal that many failures involve multiple distinct trials or ambiguous agent interactions, making single-step attribution inherently problematic. This motivates the intervention-based framing in a compelling way.
2. The paper presents a well-thought-out debugging system that is modular and scalable.
3. The authors conduct thorough experiments across three datasets with detailed quantitative and qualitative evaluations. They categorize outcomes (validated, partially validated, refuted, inconclusive), providing a clear framework to assess debugging performance.
4. The visualization system will be highly useful to researchers building or analyzing LLM agent systems. If released, the framework could become a standard tool for debugging such systems in both academia and industry.

**Weaknesses:**

1. As the authors acknowledged, while the decision to focus on orchestrator-level interventions avoids invasive modifications, it also limits the generality of DoVer. Some failures likely stem from sub-agent capabilities, which the current system cannot directly address. Probing the boundary of sub-agents and determine whether intervention would work could be a potential direction.
2. Although the system is said to be modular, it depends on magnetic-one with checkpointing and re-execution capabilities. This may raise the bar for adoption in settings where such infrastructure is not readily available.
3. The paper did not discuss [1] in depth, which is also an important taxomony for multi-agent system failures. It would be important to discuss how DoVer could be leveraged to address the failure attribution dataset proposed in [1].

[1] Why Do Multi-Agent LLM Systems Fail?

**Questions:**

Please discuss the weakness section.

---

> ### Author Response · Authors · 2025-11-21
> **Responses to Reviewer tv3u (1/2)**
>
> We sincerely appreciate your thorough review of our paper. Below are our responses to your concerns:
>
> > **W1.** As the authors acknowledged, while the decision to focus on orchestrator-level interventions avoids invasive modifications, it also limits the generality of DoVer. Some failures likely stem from sub-agent capabilities, which the current system cannot directly address. Probing the boundary of sub-agents and determine whether intervention would work could be a potential direction.
>
> Thank you for pointing out the limitations of orchestrator-level interventions. We fully agree that some failures originate from insufficient sub-agent capabilities and therefore cannot be fully resolved by orchestrator interventions alone. This is also reflected in the *inconclusive* cases reported in Table 2. Importantly, these cases are not merely failures but **valuable signals**: they delineate the *boundary* of sub-agent capabilities as you pointed out, and directly highlight where system-level improvements are needed.
>
> We therefore view DoVer as a key component of a broader debugging loop:
> (i) DoVer provides automated, orchestrator-level interventions;
> (ii) failures that remain unresolved by DoVer reveal sub-agent bottlenecks that may require targeted human or automated refinement.
>
> To illustrate the effectiveness of this “DoVer automation + human-in-the-loop” cycle, we analyzed the failure traces of inconclusive cases in WW-GAIA and discovered two recurring failure modes in **WebSurfer**: (i) **Lack of a “scroll-to-bottom” tool**, causing repetitive loops of single-page scrolling. (ii) **Inability to process PDF content**, leading to errors like “no information to summarize”. These issues directly limited the Orchestrator’s ability to recover the task. Based on these insights, we implemented two targeted improvements to WebSurfer: a “scroll-to-bottom” tool, and revised PDF-handling logic (downloading PDFs before processing instead of opening them directly in the browser).
>
> With these enhancements, Cases 20, 21, and 26, which are all previously labeled “inconclusive” because WebSurfer could not carry out required actions, can now be successfully handled by DoVer using *orchestrator-level* interventions alone. This demonstrates that DoVer not only identifies recoverable failures but also reveals concrete sub-agent bottlenecks, enabling systematic system improvement.
>
> Looking forward, we see two promising directions depending on the degree of permissible system modification: (i) **Fully automated debugging loops**, where DoVer’s insights feed into automated sub-agent improvement (e.g., using a coding agent to implement fixes). (ii) **Capability-aware intervention generation**, enabling DoVer to adapt its interventions to known sub-agent limitations in settings where agent code cannot be modified. Both directions align with the broader vision of developing robust, self-improving multi-agent systems.

---

> > ### Author Response · Authors · 2025-11-21
> > **Responses to Reviewer tv3u (2/2)**
> >
> > > **W2.** Although the system is said to be modular, it depends on magnetic-one with checkpointing and re-execution capabilities. This may raise the bar for adoption in settings where such infrastructure is not readily available.
> >
> > We appreciate the concern regarding DoVer’s reliance on checkpointing and re-execution. Indeed, many existing multi-agent frameworks do not provide this functionality out of the box. However, we found that adding checkpointing is achievable with moderate engineering effort.
> >
> > To demonstrate feasibility, we implemented a fully functional checkpointing and re-execution system for the **AG2** framework, which originally lacked such capabilities. The implementation includes capturing and restoring conversation state, agent configurations, and LLM parameters, and required ~2,000 lines of code and roughly three person-days, aided by LLM-based coding tools. We have released this extension in our anonymous repository linked in the paper.
> >
> > From this experience, we highlight several design principles that may lower adoption barriers:
> > 1. **Minimal-invasive Design**: By wrapping core components (e.g., chat manager in AG2) with checkpoint-aware versions, we preserved existing architecture while enabling checkpointing.
> > 2. **Capture Full System State**: We used LLM-assisted tooling to identify all relevant state variables (e.g., agent configs, LLM settings, conversation history), ensuring accurate restoration.
> > 3. **Intervention via Checkpoint Manipulation**: Interventions operate directly on structured checkpoint data, enabling seamless reuse of restoration and replay logic.
> >
> > We hope these implementation guidelines make it easier to integrate DoVer into a wide range of agent systems.
> >
> > > **W3.** The paper did not discuss [1] in depth, which is also an important taxomony for multi-agent system failures. It would be important to discuss how DoVer could be leveraged to address the failure attribution dataset proposed in [1]. [1] Why Do Multi-Agent LLM Systems Fail?
> >
> > Thank you for highlighting the connection to the MAST work [1]. While MAST-Data provides high-level failure labels for complete multi-agent runs, DoVer focuses on fine-grained, trial-level error attribution, so the two offer complementary views on where and how multi-agent systems fail. Following your suggestion, we evaluated DoVer on the **GSMPlus** math dataset used in [1]. We built a **MathChat** multi-agent system within the **AG2** framework using the prompts from [1], and extended the system with checkpointing and re-execution capabilities. Since DoVer requires checkpointing, the failure traces in [1] could not be used directly; instead, we re-collected execution traces with checkpoints by running all **2,400 samples** in the “testmini” split of the original GSMPlus dataset. We then applied DoVer to intervene on the failed cases, using GPT-4o as the DoVer model while powering the MathChat system with two different agent models (GPT-4o and GPT-4o-mini). The results are:
> > | Agent Model | Failed Cases | Intervened Trials | Succeed Trials | Trial Success Rate |
> > |-------------|--------------|-------------------|----------------|--------------|
> > | GPT-4o      | 214          | 198                 | 97              | 49.0 %        |
> > | GPT-4o-mini | 294         | 230                 | 111              | 48.3 %        |
> >
> > These results show that DoVer achieves **~50%** success rates on GSMPlus, extending the effectiveness observed on Who&When to a different benchmark under a different agent framework. These findings demonstrate that DoVer generalizes well across datasets and agent frameworks. We will include the complete results in the revised manuscript.

---

> > > ### Comment · Reviewer_tv3u · 2025-11-21
> > >
> > > I thank the authors for crafting such a detailed and extensive rebuttal. I see enormous engineering effort in putting together the additional experiments. I believe this work is a valuable step towards automated debugging of agentic systems and is worthy of acceptance. I have revised my score accordingly.

---

> > > > ### Author Response · Authors · 2025-11-26
> > > >
> > > > Thank you very much for your thoughtful follow-up and for taking the time to reconsider your evaluation. We sincerely appreciate your recognition of the additional experiments and engineering effort in our rebuttal. Following your suggestions, we have updated the PDF to improve clarity and completeness, and we are grateful that you found the revisions valuable!

---

### Official Review · Reviewer_hTW1 · 2025-10-31

**Soundness:** 3
**Presentation:** 3
**Contribution:** 3
**Rating:** 6
**Confidence:** 4

**Summary:**

This paper proposes DoVer, an intervention-driven framework for debugging failures in LLM-based multi-agent systems. Instead of only doing log-based failure attribution (e.g., Who&When, AgentTracer), the paper argues that such attribution is often ambiguous, particularly because: 1. Multi-agent systems have multiple trials and branching traces 2. Inter-agent coordination errors exist 3. Ground-truth annotation is noisy and uncertain. To address this, DoVer generates failure hypotheses from logs, actively intervenes (e.g., modifying orchestrator instructions or plans), replays the task from that checkpoint, and evaluates whether the failure is fixed or progress is made. Experiments on AssistantBench and GAIA show DoVer can flip 18–28% failed tasks to success and validate/refute 30–60% hypotheses.

**Strengths:**

1. Originality: Creative shift from log-only evaluation to active debugging paradigm.
2. I like the idea of the Insightful finding that ground-truth failure labels are inherently ambiguous in multi-agent setting; this is very valuable observation, e.g., “multiple trials per session and inter-agent misalignment make single-step annotation ill-posed”.
3. Trial segmentation idea and use of checkpoint replay is practical and generalizable.
4. Instead of just pointing at logs, they actually intervene, modify instructions, and rerun. This is very realistic. If you are real agent engineer you know you always try “what if we change instruction here?” And surprisingly they get ~18–28% failure flipped to success. This is quite good considering agent tasks are messy

**Weaknesses:**

1. Although they use realistic benchmarks, dataset size is not huge (~100 cases, ~200 trials). I worry behavior may differ when: tasks have interactive state, not only web browsing, open source weaker agents used (GPT-4o/5 are strong babysitters)
2. All experiments on Magentic-One + AGDebugger. What about the agents from other frameworks? Would trial segmentation generalize?

**Questions:**

1. How would DoVer handle very long traces (hundreds of steps)? Is trial segmentation always robust?
2. Another open question is: Can intervention generation be learned? For example, RL finetuning to suggest optimal edits rather than LLM heuristics?

---

> ### Author Response · Authors · 2025-11-21
> **Responses to Reviewer hTW1 (1/2)**
>
> We sincerely appreciate your thorough review of our paper. Below are our responses to your concerns:
>
> > **W1.** Although they use realistic benchmarks, dataset size is not huge (~100 cases, ~200 trials). I worry behavior may differ when: tasks have interactive state, not only web browsing, open source weaker agents used (GPT-4o/5 are strong babysitters)
>
> Thank you for highlighting the importance of evaluating DoVer on more diverse datasets and agent configurations. In response, we expanded our evaluation in two complementary directions.
>
> **(A) Additional Dataset: GSMPlus Math Reasoning**: Following your suggestion and Reviewers wRjk/tv3u’s feedback, we evaluated DoVer on the **GSMPlus** math dataset used in the recent multi-agent failure-taxonomy study [1], under a different agent framework, **AG2**. We built a **MathChat** multi-agent system within AG2 using the prompts from [1], and extended the system with checkpointing and re-execution capabilities. Since DoVer requires checkpointing, the failure traces in [1] could not be used directly; instead, we re-collected execution traces with checkpoints by running all **2,400 samples** in the “testmini” split of the original GSMPlus dataset. We then applied DoVer to intervene on the failed cases, using GPT-4o as the DoVer model while powering the MathChat system with two different agent models (GPT-4o and GPT-4o-mini). The results are:
> | Agent Model | Failed Cases | Intervened Trials | Succeed Trials | Trial Success Rate |
> |-------------|--------------|-------------------|----------------|--------------|
> | GPT-4o      | 214          | 198                 | 97              | 49.0 %        |
> | GPT-4o-mini | 294         | 230                 | 111              | 48.3 %        |
>
> These results show that DoVer achieves **~50%** success rates on GSMPlus, extending the effectiveness observed on Who&When to a different benchmark under a different agent framework.
>
> **(B) Additional Models: Locally Hosted Open-Source Agents**: Following your feedback and the related suggestion from Reviewer wRjk, we evaluated DoVer with two open-source models (**Qwen3-8B** and **Qwen3-32B**) both run locally with thinking mode enabled. We evaluated DoVer on the WW-GAIA dataset using the same failure traces and prompts as in the GPT-4o experiments, and performed **three** independent runs per model to reduce variance. Below are the results:
> | DoVer Model           | Intervened Trials | Avg. Trial Success Rate|
> |-----------------------|-------------------|------------------------|
> | Qwen3-8B     | 76                | 11.0 %                |
> | Qwen3-32B    | 87                | 16.9 %                |
> | GPT-4o       | 99                | 17.6 %                |
>
> The results reveal that: **(i) DoVer is effective across different LLM architectures**, achieving meaningful success rates with both 8B- and 32B-sized open-source models; **(ii) Larger models tend to yield better intervention outcomes**, as evidenced by the higher success rate of Qwen3-32B compared to Qwen3-8B; **(iii) Qwen3-32B performs comparably to GPT-4o**, showing that high-quality open-source models can support DoVer effectively, enhancing reproducibility and accessibility.
>
> Together, these results show that DoVer generalizes well across datasets, model scales, and agent systems. We will incorporate the full results into the revised manuscript.
>
> [1] Cemri, M., Pan, M., Yang, S., et al. (2025). Why Do Multi-Agent LLM Systems Fail?
>
>
> > **W2.** All experiments on Magentic-One + AGDebugger. What about the agents from other frameworks? Would trial segmentation generalize?
>
> Thank you for raising the important question of DoVer’s generalizability across different agent frameworks. As shown in our response to W1, we have successfully applied DoVer to a multi-agent system built in the **AG2** framework, demonstrating that DoVer does generalize beyond Magentic-One.
>
> However, we do notice architectural differences between Magentic-One and MathChat that lead to different trial segmentation characteristics. Unlike Magentic-One’s explicit Planner–Executor loop, MathChat uses a group-chat architecture consisting of a manager and three specialized agents (ProblemSolver, CodeExecutor, Verifier). Agents respond when selected by the manager based on conversation context. Although there is no explicit planner as Orchestrator in Magentic-One, we observe group-level divergent trial paths initiated by different specialized agents (e.g., a new problem-solving attempt by ProblemSolver or a new checking cycle by Verifier). The number of trials is smaller in MathChat on GSMPlus (about 1.3 per case), but trials remain clearly distinguishable.

---

> > ### Author Response · Authors · 2025-11-21
> > **Responses to Reviewer hTW1 (2/2)**
> >
> > > **Q1.** How would DoVer handle very long traces (hundreds of steps)? Is trial segmentation always robust?
> >
> > We appreciate the question regarding scalability. In our current datasets (WW and GSMPlus), we have *not* yet encountered extremely long traces; the longest trace observed is ~100 steps, for which segmentation remains accurate.
> >
> > For traces with hundreds of steps, which may arise in longer-horizon tasks, we foresee potential challenges including potential context-window limits and greater structural complexity. However, we believe trial segmentation remains tractable for two reasons: (i) Trial boundaries typically correspond to planning or strategy-update turns that depend primarily on local conversational context rather than long-range dependencies. This makes the segmentation task inherently robust even as trace length grows. (ii) If traces exceed the model’s context window, we can process them using a sliding-window approach: the trace is divided into overlapping segments that individually fit within the context limit. Segmentation is performed on each segment, and the local predictions are then aggregated to produce a consistent global segmentation.
> >
> > Overall, while extremely long traces introduce practical considerations, the largely local nature of trial boundaries and the feasibility of sliding-window segmentation suggest that DoVer can scale effectively. We will add a discussion of this point to the revised manuscript.
> >
> > > **Q2.** Another open question is: Can intervention generation be learned? For example, RL finetuning to suggest optimal edits rather than LLM heuristics?
> >
> > Thank you for raising the intriguing possibility of learning-based intervention generation. We explored this direction by fine-tuning smaller models (e.g., Qwen3-8B), but found that we currently lack sufficient training data to achieve that. As an alternative, we investigated few-shot prompting, which offered a more data-efficient way to guide model behavior.
> >
> > Specifically, we evaluated Qwen3-8B with three few-shot examples designed to better guide its intervention generation. The results, together with the baseline 0-shot result, are shown below:
> > | DoVer Model           | Intervened Trials | Avg. Trial Success Rate|
> > |-----------------------|-------------------|------------------------|
> > | Qwen3-8B (0-shot)     | 76                | 11.0 %                |
> > | Qwen3-8B (**3-shot**) | 77                | **14.3 %**            |
> >
> > These results indicate that few-shot prompting can substantially improve performance for smaller models: Qwen3-8B reaches a 14.3% success rate with 3-shot prompting, outperforming its own 0-shot baseline. Looking ahead, further gains may come from improved prompting strategies (e.g., additional examples, chain-of-thought guidance). At the same time, collecting larger-scale intervention data could eventually enable effective supervised fine-tuning or reinforcement-learning–based intervention-generation models. We will incorporate this discussion into the revised manuscript.

---

> ### Author Response · Authors · 2025-11-26
> **Follow-up on Rebuttal**
>
> Dear Reviewer hTW1,
>
> Thank you again for your thoughtful and constructive review. We have prepared a rebuttal and updated the revised PDF to address your main concerns. As the discussion period ends on December 2, we would greatly appreciate it if you could briefly review our responses and let us know whether they resolve your concerns or if any points remain unclear. We are happy to provide further clarification if needed.
>
> The authors of DoVer

---

> > ### Comment · Reviewer_hTW1 · 2025-11-26
> > **Response**
> >
> > Thanks to the authors for the rebuttal. I will keep my score. My concerns are addressed.

---

### Official Review · Reviewer_ZnZ9 · 2025-11-01

**Soundness:** 2
**Presentation:** 2
**Contribution:** 2
**Rating:** 4
**Confidence:** 3

**Summary:**

This work introduces DoVer, an intervention-driven debugging framework for multi-agent systems. The motivation lies in that 1) existing log-only failure attribution cannot validate its hypothesis and 2) single-step or single-agent attribution (as assessed by existing benchmarks) is often inherently ill-posed because of the multi-trial nature of agentic systems. On constructed benchmark, the proposed DoVer could flip 18-28% of failed trials into successes.

**Strengths:**

The concrete and detailed motivating analysis of log-based failure attribution in Section 3 is quite informative in exposing how existing benchmarks and metrics could be insufficient and/or inaccurate for evaluating agent failure attribution.

**Weaknesses:**

My major concern is that there is no baseline in current experiments, making it really difficult to interpret how good the numbers in Table 2 are and the proposed method is.

Specifically, the proposed intervention-based debugging system is essentially doing self-refinement in some sense. Therefore, I believe reasonable baselines could include self-improvement techniques such as Self-Refine (Madaan et al., 2023) and CRITIC (Gou et al., 2024) which provide feedbacks to the agentic system to improve its performance in the next round.

**Questions:**

N/A

---

> ### Author Response · Authors · 2025-11-21
> **Responses to Reviewer ZnZ9**
>
> We sincerely appreciate your thorough review of our paper. Below are our responses to your concerns:
>
> > **W1.** My major concern is that there is no baseline in current experiments, making it really difficult to interpret how good the numbers in Table 2 are and the proposed method is. Specifically, the proposed intervention-based debugging system is essentially doing self-refinement in some sense. Therefore, I believe reasonable baselines could include self-improvement techniques such as Self-Refine (Madaan et al., 2023) and CRITIC (Gou et al., 2024) which provide feedbacks to the agentic system to improve its performance in the next round.
>
> Thank you for your valuable suggestion regarding the inclusion of self-improvement baselines. While Self-Refine and CRITIC were not originally designed for multi-agent settings, we adapted their core mechanisms to create two baselines:
> 1. **Self-Refine-style Baseline**: Self-Refine asks a model to critique and improve its own output. In our adaptation, for each failure case, we prompt the underlying LLM (GPT-4o) to: (i) generate feedback based on the system’s final answer and full session history, and (ii) produce a revised final answer using this feedback. Both prompts follow the structure of the original Self-Refine framework, with minimal modifications to suit our multi-agent context.
> 2. **CRITIC-style Baseline**: CRITIC leverages external feedback to guide iterative improvement. In our adaptation, we again ask the LLM to produce feedback from the final answer and session history. Instead of directly revising the answer, we insert this feedback as a new agent message into the conversation trace. This allows Magentic-One to resume the conversation for an additional round, enabling all agents, such as those that can query external tools (e.g., WebSurfer, CodeExecutor), to jointly produce an improved answer.
>
> We evaluated both baselines on the WW-GAIA dataset using the exact same underlying model (GPT-4o) and the same failure traces used in the DoVer experiments. Across all 26 failed cases, both baselines achieved a **0%** recovery rate, failing to flip any unsuccessful case into a successful one. In contrast, DoVer achieves an **18%** success rate at the trial level, as reported in Table 2.
>
> A qualitative examination reveals why these self-improvement baselines fall short: trial trajectories between the initial failure point and the final answer are often long, noisy, and highly divergent, making end-of-trace refinement insufficient for reliably redirecting the system. In contrast, DoVer performs *in-situ* interventions at potential failure points, enabling timely and targeted corrections that are essential in multi-agent settings.
>
> We will incorporate these baseline implementations, results, and analyses into the revised manuscript to provide clearer comparative context and to strengthen the empirical justification for DoVer’s effectiveness.

---

> ### Author Response · Authors · 2025-11-26
> **Follow-up on Rebuttal**
>
> Dear Reviewer ZnZ9,
>
> Thank you again for your thoughtful and constructive review. We have prepared a rebuttal and updated the revised PDF to address your main concerns. As the discussion period ends on December 2, we would greatly appreciate it if you could briefly review our responses and let us know whether they resolve your concerns or if any points remain unclear. We are happy to provide further clarification if needed.
>
> The authors of DoVer

---

### Official Review · Reviewer_wRjk · 2025-11-01

**Soundness:** 2
**Presentation:** 3
**Contribution:** 2
**Rating:** 4
**Confidence:** 5

**Summary:**

DoVer is a framework that treats debugging in LLM-based multi-agent systems as an intervention-driven “do-then-verify” process rather than passive log analysis. It systematically tests and validates failure hypotheses through targeted interventions, recovering up to 28% of failed trials and confirming or disproving most hypotheses.

**Strengths:**

The in-depth examination of the Who&When dataset in Section 3 provides valuable insights into the system’s behavior, helping clarify when and why failures occur.

The framework’s ability to transform failed trials into useful learning opportunities demonstrates its robustness and practical value for improving multi-agent reliability.

**Weaknesses:**

The evaluation relies solely on a hand-crafted subset of the Who&When dataset, which limits the generalizability of the findings and raises questions about how well DoVer would perform on larger or more diverse real-world datasets.

The framework is evaluated only with GPT-4o as the backend model, which restricts understanding of DoVer’s effectiveness across different LLM architectures and limits claims about its general applicability.

**Questions:**

Some citation format is incorrect, eg line 151 React citation format is wrong

It would strengthen the work to train and evaluate a smaller, locally hosted model for the task, as this would demonstrate DoVer’s adaptability beyond proprietary large models and improve reproducibility and accessibility for broader research use.

---

> ### Author Response · Authors · 2025-11-21
> **Responses to Reviewer wRjk (1/2)**
>
> We sincerely appreciate your thorough review of our paper. Below are our responses to your concerns:
>
> > **W1.** The evaluation relies solely on a hand-crafted subset of the Who&When dataset, which limits the generalizability of the findings and raises questions about how well DoVer would perform on larger or more diverse real-world datasets.
>
> Thank you for raising the concern regarding the dataset scope and the generalizability of DoVer. Following your feedback and the related suggestion from Reviewers hTW1/tv3u, we extended our evaluation to the **GSMPlus** math dataset used in the recent multi-agent failure-taxonomy study [1], under a different multi-agent framework, **AG2**.
>
> Specifically, we constructed a **MathChat** multi-agent system using the prompts from [1] within the AG2 framework. Because DoVer requires checkpointing, and the traces released in [1] do not contain checkpoints, we augmented AG2 with checkpointing and re-execution capabilities (analogous to AGDebugger for Magentic-One) and re-collected execution traces with checkpoints by running all **2,400 samples** in the “testmini” split of the original GSMPlus dataset. We then applied DoVer to intervene on the failed cases, using GPT-4o as the DoVer model while powering the MathChat system with two different agent models (GPT-4o and GPT-4o-mini). The results are:
> | Agent Model | Failed Cases | Intervened Trials | Succeed Trials | Trial Success Rate |
> |-------------|--------------|-------------------|----------------|--------------|
> | GPT-4o      | 214          | 198                 | 97              | 49.0 %        |
> | GPT-4o-mini | 294         | 230                 | 111              | 48.3 %        |
>
> These results show that DoVer achieves **~50%** success rates on GSMPlus, extending the effectiveness observed on Who&When to a different benchmark under a different agent framework. These findings demonstrate that DoVer generalizes well across datasets and agent frameworks. We will include the complete results in the revised manuscript.
>
> [1] Cemri, M., Pan, M., Yang, S., et al. (2025). Why Do Multi-Agent LLM Systems Fail?

---

> > ### Author Response · Authors · 2025-11-21
> > **Responses to Reviewer wRjk (2/2)**
> >
> > > **W2.** The framework is evaluated only with GPT-4o as the backend model, which restricts understanding of DoVer’s effectiveness across different LLM architectures and limits claims about its general applicability. \
> > > **Q2.** It would strengthen the work to train and evaluate a smaller, locally hosted model for the task, as this would demonstrate DoVer’s adaptability beyond proprietary large models and improve reproducibility and accessibility for broader research use.
> >
> > Thank you for emphasizing the importance of evaluating DoVer across diverse LLM architectures, including smaller and locally hosted models. To address this, we conducted additional experiments in two complementary directions:
> > 1. **Open-Source Models as Backend**: We evaluated DoVer with two open-source models (**Qwen3-8B** and **Qwen3-32B**) both run locally with thinking mode enabled. To ensure comparability, we used the same prompts as in our GPT-4o experiments (without few-shot examples).
> > 2. **Few-Shot Prompting for Smaller Models**: As you suggested, training a smaller DoVer model would be desirable. However, we currently lack sufficient training data to effectively fine-tune such a model. As a practical alternative, we explored whether few-shot prompting could boost the effectiveness of a smaller model. In particular, we evaluated Qwen3-8B with three few-shot examples, designed to better guide its intervention generation.
> >
> > We evaluated the above settings on the WW-GAIA dataset using the same failure traces as in the GPT-4o experiments, performing **three** independent runs per setting to reduce variance. The aggregated results are shown below:
> > | DoVer Model           | Intervened Trials | Avg. Trial Success Rate|
> > |-----------------------|-------------------|------------------------|
> > | Qwen3-8B (0-shot)     | 76                | 11.0 %                |
> > | Qwen3-32B (0-shot)    | 87                | 16.9 %                |
> > | GPT-4o (0-shot)       | 99                | 17.6 %                |
> > | Qwen3-8B (**3-shot**) | 77                | **14.3 %**            |
> >
> > The results reveal that: **(i) DoVer is effective across different LLM architectures**, achieving meaningful success rates with both 8B- and 32B-sized open-source models; **(ii) Larger models tend to yield better intervention outcomes**, as evidenced by the higher success rate of Qwen3-32B compared to Qwen3-8B; **(iii) Qwen3-32B performs comparably to GPT-4o**, showing that high-quality open-source models can support DoVer effectively, enhancing reproducibility and accessibility; **(iv) Few-shot prompting significantly boosts performance for smaller models**, with Qwen3-8B achieving a 14.3% success rate with 3-shot examples, surpassing its 0-shot performance.
> >
> > We will include these results in the revised manuscript to strengthen claims about DoVer’s generality, its accessibility for open-source deployment, and the potential of few-shot prompting to enhance smaller models, while leaving room for future work on training specifically tailored smaller models.
> >
> > > Q1. Some citation format is incorrect, eg line 151 React citation format is wrong.
> >
> > Thank you for pointing out the citation formatting error (e.g., line 151). We have corrected the format and ensured consistency throughout the revised manuscript.

---

> ### Author Response · Authors · 2025-11-26
> **Follow-up on Rebuttal**
>
> Dear Reviewer wRjk,
>
> Thank you again for your thoughtful and constructive review. We have prepared a rebuttal and updated the revised PDF to address your main concerns. As the discussion period ends on December 2, we would greatly appreciate it if you could briefly review our responses and let us know whether they resolve your concerns or if any points remain unclear. We are happy to provide further clarification if needed.
>
> The authors of DoVer

---

> > ### Comment · Reviewer_wRjk · 2025-11-27
> > **Rebuttal looks good**
> >
> > The additional results look good to me, I would suggest add more datasets and open models in your next version.

---

### Author Response · Authors · 2025-12-02
**Note to AC: Rebuttal Summary & Acknowledgement**

Dear Area Chair,

We would like to begin by expressing our sincere gratitude for your willingness to take on our submission under these unusual circumstances and for the time and care you are devoting to its evaluation. We are also deeply grateful to all reviewers (**wRjk, ZnZ9, hTW1, tv3u**) and to you as AC for your thoughtful, constructive feedback; below we provide a summary of our responses during the rebuttal phase.

---

### Strength
Across the four reviews, the following strengths were consistently highlighted:
- Clear and insightful analysis of the limitations of log-based failure attribution in multi-agent systems (e.g., ambiguity of single-step labels, importance of trials and inter-agent misalignment). (**wRjk, ZnZ9, hTW1, tv3u**)
- A realistic and creative shift from passive log evaluation to active, intervention-based debugging. (**hTW1, tv3u**)
- Practical design of trial segmentation and checkpoint-based replay, which matches how real agent engineers debug systems. (**hTW1**)
- A modular, scalable debugging system (DoVer) that performs in-situ interventions and flips a non-trivial fraction of failures to successes on realistic, messy tasks. (**wRjk, hTW1, tv3u**)
- Thorough quantitative and qualitative evaluations (including outcome categories like validated/partially validated/refuted/inconclusive) and a visualization/debugging tool that could become a standard resource for the community. (**tv3u**)

---

### Summary of Our Rebuttal
In the following, we organize the concerns from reviewers and our rebuttal in terms of several themes:

#### T1: Method Generalization over Different Datasets and Frameworks

- Sources of Reviewers’ Concerns: **wRjk (W1), hTW1 (W1/W2), tv3u (W2/W3)**
- To address concerns about generalization across datasets and frameworks, we extended our evaluation to the **GSMPlus** math dataset from [1], as suggested by Reviewer tv3u, and used a different agent framework, **AG2**.
- Implementation Details:
  (i) Implemented checkpointing and re-execution for AG2 (≈2,000 LOC; ~3 person-days), capturing conversation state, agent configs, and LLM parameters, and enabling interventions via structured checkpoint manipulation;
  (ii) Built a MathChat multi-agent system in AG2 using the prompts from [1];
  (iii) Re-collected execution traces with checkpoints by running all **2,400** samples in the GSMPlus “testmini” split. We evaluated two agent models: GPT-4o and GPT-4o-mini.
- Experimental Results (DoVer on failed GSMPlus cases):

  | Agent Model | Failed Cases | Intervened Trials | Succeed Trials | Trial Success Rate |
  |------------|-------------:|------------------:|---------------:|-------------------:|
  | GPT-4o     |          214 |               198 |             97 |              49.0% |
  | GPT-4o-mini|          294 |               230 |            111 |              48.3% |

- Implications:
  (i) DoVer attains ≈50% trial success on GSMPlus, extending the effectiveness observed on Who&When to a different benchmark and domain under a different agent framework;
  (ii) Our rapid implementation of checkpointing and replay for AG2 shows that DoVer can be ported beyond Magentic-One/AGDebugger with moderate engineering effort;
  (iii) Trial segmentation transfers to AG2’s group-chat architecture (manager + ProblemSolver, CodeExecutor, Verifier), where we still observe clearly distinguishable trials (≈1.3 per case).

[1] Cemri, M., Pan, M., Yang, S., et al. (2025). Why Do Multi-Agent LLM Systems Fail?

---

> ### Author Response · Authors · 2025-12-02
> **Note to AC: Rebuttal Summary & Acknowledgement (2)**
>
> #### T3: Comparison with Other Self-Improvement Baselines
>
> - Sources of Reviewers’ Concerns: **ZnZ9 (W1)**
> - To address concerns that our method resembles self-refinement and lacks appropriate baselines (e.g., Self-Refine, CRITIC), we adapted the core mechanisms of these self-improvement methods to the multi-agent setting and directly compared them with DoVer.
> - Implementation Details:
>   - Self-Refine asks a model to critique and improve its own output. In our adaptation, for each failure case, we prompt the underlying LLM (GPT-4o) to (i) generate feedback based on the system’s final answer and full session history, and (ii) produce a revised final answer using this feedback, following the structure of the original Self-Refine framework with minimal modifications to suit our multi-agent context.
>   - CRITIC leverages external feedback to guide iterative improvement. In our adaptation, we again ask the LLM to produce feedback from the final answer and session history; instead of directly revising the answer, we insert this feedback as a new agent message into the conversation trace, then allow Magentic-One to resume the conversation for an additional round so all agents, including tool-using ones such as WebSurfer and CodeExecutor, can jointly produce an improved answer.
>   - Both baselines were evaluated on the WW-GAIA dataset using the exact same underlying model (GPT-4o) and the same failure traces as in the DoVer experiments (26 failed cases).
> - Experimental Results: Across all 26 failed cases, both baselines achieved a 0% recovery rate, failing to flip any unsuccessful case into a successful one. In contrast, DoVer achieves an 18% success rate at the trial level on the same set, as reported in Table 2.
> - Analysis and Implication: A qualitative examination reveals why these self-improvement baselines fall short: trial trajectories between the initial failure point and the final answer are often long, noisy, and highly divergent in multi-agent systems, making end-of-trace refinement insufficient for reliably redirecting behavior. In contrast, DoVer performs in-situ interventions at potential failure points via checkpointed replay, enabling timely and targeted corrections that are essential in multi-agent settings and substantially more effective than these adapted self-improvement baselines.
>
> #### T4: Probing Sub-Agent Capability Boundaries
>
> - Sources of Reviewers’ Concerns: **tv3u (W1)**
> - To address concerns that orchestrator-level interventions may miss failures rooted in sub-agent capabilities, we showed how DoVer helps reveal and probe the boundaries of sub-agent/tool capabilities, and how this enables a practical debugging loop.
> - Implementation Details and Findings: On WW-GAIA, we examined “inconclusive” DoVer cases and identified recurring WebSurfer limitations: a missing “scroll-to-bottom” tool, which caused repetitive single-page scrolling loops, and an inability to process PDF content, leading to errors like “no information to summarize.” Guided by these signals, we added a “scroll-to-bottom” tool to WebSurfer and revised PDF handling so that WebSurfer downloads PDFs before processing them, rather than opening them directly in the browser.
> - Results and Implications: After these targeted improvements, previously inconclusive Cases 20, 21, and 26 became recoverable using orchestrator-level DoVer interventions alone. This illustrates a practical debugging loop: DoVer’s orchestrator-level interventions diagnose where sub-agent/tool capabilities are insufficient, and focused upgrades to those components expand the range of failures that orchestration can successfully recover. Thus, even though DoVer operates at the orchestrator level, it naturally exposes sub-agent boundaries and guides capability improvements.

---

> ### Author Response · Authors · 2025-12-02
> **Note to AC: Rebuttal Summary & Acknowledgement (3)**
>
> #### T5: Method Robustness for Handling Very Long Traces and Trial Segmentation
>
> - Sources of Reviewers’ Concerns: **hTW1 (Q1/W2)**
> - To address concerns about very long traces and segmentation robustness, we clarified the empirical range of our traces and discussed why trial segmentation remains tractable for longer horizons and different frameworks.
> - Clarification details:
>   - In our current datasets (Who&When and GSMPlus), the longest trace is ≈100 steps, and trial segmentation is accurate in this regime.For much longer traces (hundreds of steps), trial boundaries are typically determined by local planning/strategy-update turns that mainly depend on local context, which helps segmentation remain robust as traces grow. If a trace exceeds the model’s context window, we can use a sliding-window approach: split it into overlapping segments, segment each locally, and aggregate the results to obtain consistent global boundaries.
>   - Trial segmentation also transfers to AG2’s group-chat architecture (manager + ProblemSolver, CodeExecutor, Verifier), where we still observe clear trials (≈1.3 per case) corresponding to strategy shifts (e.g., new ProblemSolver attempts or Verifier cycles). While ultra-long traces were not present in our current benchmarks, the local nature of trial boundaries and the feasibility of sliding-window processing suggest that DoVer’s segmentation can scale to longer-horizon tasks and different agent architectures.
>
> ---
>
> ### Reviewers’ Follow-up during Rebuttal
>
> | Reviewer |Confidence |  Main Themes Addressed | Follow-up After Rebuttal (Before Score Rollback) |
> |---------|---------|------------------------|--------------------------------------------------|
> | wRjk  | 5  | T1 (datasets/frameworks), T2 (models) | Confirmed that the added GSMPlus/AG2 experiments and open-source model results look good to him/her and raised the score from 4 to 6. |
> | ZnZ9   | 3 | T3 (baselines)         | No reply.|
> | hTW1  | 4  | T1 (generalization), T2 (learning/prompting), T5 (long traces/segmentation) | Maintained a positive rating of 6, acknowledging that the extended evaluation results and our clarification on long trace and segmentation robustness addressed his/her concerns. |
> | tv3u  | 5  | T1 (datasets/frameworks), T4 (sub-agent boundaries) | Appreciated our additional experiments on GSMPlus/AG2 and tool-enhancement, and commented that this work can be a valuable step towards automated debugging of agentic systems. Recommended for acceptance and raised the score from 6 to 8.|
>
> We believe these updates and clarifications directly address the main concerns raised by Reviewers **wRjk, ZnZ9, hTW1**, and **tv3u**, and further strengthen the case for DoVer as a robust, general, and practical framework for debugging multi-agent LLM systems. We hope this overview is helpful for your final assessment.
>
> Best regards,
>
> The Authors

---

### Meta-Review · Area_Chair_2hKr · 2026-01-18

**Summary:**

This paper proposes DoVer, an intervention-driven framework for debugging failures in LLM-based multi-agent systems. Instead of relying solely on log-based failure attribution, DoVer actively generates and verifies failure hypotheses through targeted interventions and checkpointed replay, evaluating success based on task recovery or measurable progress. Reviewers broadly agreed that the problem is important and timely, and highlighted the shift from passive attribution to outcome-oriented debugging as a meaningful and practical contribution.

**Reviewer Concerns:**

The main concerns raised in the initial reviews centered on generalization, baseline comparisons, and system dependence. Some reviewers questioned whether results obtained on specific benchmarks and agent frameworks would transfer to other datasets, models, or architectures. Others requested clearer comparisons to self-improvement baselines such as Self-Refine and CRITIC, and raised questions about reliance on checkpointing infrastructure and orchestrator-level interventions.

The rebuttal directly addressed these points by extending the evaluation to additional datasets and agent frameworks, adding adapted self-improvement baselines, and providing detailed analyses of long-trace handling, trial segmentation, and sub-agent capability boundaries. Reviewers who engaged after the rebuttal explicitly noted that their concerns were addressed, highlighting the added experiments, broader model coverage, and practical insights into debugging workflows.

**Reviewer Scores:**

Reviewer wRjk (initial score: 4) indicated that the additional datasets and open-model experiments looked good and raised their score to 6 prior to rollback, suggesting a positive reassessment.

Reviewer hTW1 (initial score: 6) stated that their concerns were addressed by the rebuttal and maintained a positive assessment.

Reviewer tv3u (initial score: 6) explicitly recommended acceptance after the rebuttal and increased their score to 8, citing the added experiments and engineering effort.

Reviewer ZnZ9 (initial score: 4) did not reengage after the rebuttal, but their main concern about missing baselines was directly addressed through new comparisons included by the authors.

Taken together, the post-rebuttal feedback indicates an overall upward shift in confidence among engaged reviewers.

---

### Decision · Program_Chairs · 2026-01-26

Accept (Poster)